# Chondrule-like objects and Ca-Al-rich inclusions in Ryugu may potentially be the oldest Solar System materials

Daisuke Nakashima ®[1] ✉, Tomoki Nakamura[1], Mingming Zhang[2], Noriko T. Kita ®[2], Takashi Mikouchi[3], Hideto Yoshida[4], Yuma Enokido[1], Tomoyo Morita[1], Mizuha Kikuiri[1], Kana Amano ®[1], Eiichi Kagawa[1], Toru Yada ®[5], Masahiro Nishimura[5], Aiko Nakato[5], Akiko Miyazaki ®[5], Kasumi Yogata ®[5], Masanao Abe[5], Tatsuaki Okada ®[5], Tomohiro Usui ®[5], Makoto Yoshikawa[5], Takanao Saiki[5], Satoshi Tanaka[5], Satoru Nakazawa ®[5], Fuyuto Terui[6], Hisayoshi Yurimoto ®[7], Takaaki Noguchi ®[8], Hikaru Yabuta ®[9], Hiroshi Naraoka ®[10], Ryuji Okazaki[10], Kanako Sakamoto ®[5], Sei-ichiro Watanabe ®[11], Shogo Tachibana ®[4] & Yuichi Tsuda[5]

Chondrule-like objects and Ca-Al-rich inclusions (CAIs) are discovered in the retuned samples from asteroid Ryugu. Here we report results of oxygen isotope, mineralogical, and compositional analysis of the chondrule-like objects and CAIs. Three chondrule-like objects dominated by Mg-rich olivine are $^{16}$O-rich and -poor with $\Delta^{17}O$ ($=\delta^{17}O - 0.52 \times \delta^{18}O$) values of ~ −23‰ and ~ −3‰, resembling what has been proposed as early generations of chondrules. The $^{16}$O-rich objects are likely to be melted amoeboid olivine aggregates that escaped from incorporation into $^{16}$O-poor chondrule precursor dust. Two CAIs composed of refractory minerals are $^{16}$O-rich with $\Delta^{17}O$ of ~ −23‰ and possibly as old as the oldest CAIs. The discovered objects (<30 μm) are as small as those from comets, suggesting radial transport favoring smaller objects from the inner solar nebula to the formation location of the Ryugu original parent body, which is farther from the Sun and scarce in chondrules. The transported objects may have been mostly destroyed during aqueous alteration in the Ryugu parent body.

Chondrules, Ca-Al-rich inclusions (CAIs), and fine-grained matrix are the main components of chondritic meteorites (chondrites) coming from undifferentiated asteroids[1]. Chondrules are igneous spherules composed mainly of olivine, pyroxene, glass, and Fe-Ni metal and considered to have formed by transient heating and rapid cooling[2] ~2–4 Myr after CAIs[3]. Based on the Mg# (=molar [MgO]/[MgO+FeO]%) of mafic silicates, chondrules are classified into type I (FeO-poor; Mg# ≥90) and type II (FeO-rich; Mg# <90)[2]. The Mg# of chondrules are

[1]Department of Earth Science, Tohoku University, Sendai, Miyagi 980-8578, Japan. [2]Department of Geoscience, University of Wisconsin-Madison, Madison, WI 53706, USA. [3]The University Museum, University of Tokyo, Tokyo 113-0033, Japan. [4]Department of Earth and Planetary Science, University of Tokyo, Tokyo 113-0033, Japan. [5]Institute of Space and Astronautical Science (ISAS), Japan Aerospace Exploration Agency (JAXA), Sagamihara, Kanagawa 252-5210, Japan. [6]Kanagawa Institute of Technology, Atsugi, Kanagawa 243-0292, Japan. [7]Department of Natural History Sciences, Hokkaido University, Sapporo, Hokkaido 060-0810, Japan. [8]Division of Earth and Planetary Sciences, Kyoto University, Kyoto 606-8502, Japan. [9]Department of Earth and Planetary Systems Science, Hiroshima University, Higashi-Hiroshima, Hiroshima 739-8526, Japan. [10]Department of Earth and Planetary Sciences, Kyushu University, Fukuoka 819-0395, Japan. [11]Department of Earth and Environmental Sciences, Nagoya University, Nagoya 464-8601, Japan. ✉e-mail: dnaka@tohoku.ac.jp

controlled by the oxygen fugacity of the chondrule-forming environment[4], and type I chondrules formed under more reducing conditions than type II chondrules[5]. CAIs, composed of Ca-Al-rich minerals including spinel, melilite, perovskite, hibonite, diopside, and anorthite, are condensation products in a gas of approximately solar composition near the Sun[6] or planet-forming regions at ~1 au[7] and the oldest solids in our Solar System with the U-corrected Pb-Pb absolute age of 4567.3 Ma[8,9]. A subset of CAIs experienced melting processes[6]. Amoeboid olivine aggregates (AOAs), which are lower temperature condensates than minerals constituting CAIs, consist of Mg-rich olivine, Fe-Ni metal, and Ca-Al-rich minerals including spinel, diopside, and anorthite; they are as old as CAIs[10,11]. Since chondrule-like and CAI-like objects were observed in cometary samples such as particles returned from comet Wild 2[12,13] and anhydrous interplanetary dust particles (IDPs)[14,15], it is considered that chondrules and CAIs were widely distributed from the inner Solar System to the Kuiper belt regions. Thus, chondrules and CAIs are essential for understanding of the material evolution in the early Solar System.

Oxygen-isotope ratios ($^{18}O/^{16}O$ and $^{17}O/^{16}O$) of extraterrestrial materials are known to show a wide variation, and many of them plot generally along the PCM (primitive chondrule mineral) line[16] in the oxygen three-isotope diagram, in which $^{18}O/^{16}O$ and $^{17}O/^{16}O$ ratios are converted to $\delta^{18}O$ and $\delta^{17}O$ (per mil deviations from Vienna Standard Mean Ocean Water). The $\delta^{18}O$ and $\delta^{17}O$ values of multiple mineral phases in individual chondrules from primitive chondrites (petrologic type $\leq$ 3.0) are indistinguishable within the uncertainty, except for relict grains with distinct values[16]. The homogeneous oxygen-isotope ratios represent oxygen-isotope ratios of chondrule-forming regions. Chondrules from carbonaceous chondrites have $\delta^{18}O$ and $\delta^{17}O$ values plotting along the PCM line with $\Delta^{17}O$ (=$\delta^{17}O - 0.52 \times \delta^{18}O$) ranging from ~ −5‰ to +5‰ in the oxygen three-isotope diagram, and those from ordinary chondrites have $\delta^{18}O$ and $\delta^{17}O$ values plotting above the terrestrial fractionation line with $\Delta^{17}O$ of ~ +1‰[5,17]. CAIs and AOAs generally have $^{16}O$-rich isotopic ratios with $\Delta^{17}O$ of ~ −24‰, which are nearly as $^{16}O$-rich as that of the Sun[6,18]. Relict grains occasionally found in chondrules are generally more $^{16}O$-rich ($\Delta^{17}O$ down to ~ −24‰) than coexisting mineral phases, so that the genetic link of relict grains to CAIs and AOAs has been suggested[19–21].

CI (Ivuna-type) carbonaceous chondrites consist mainly of phyllosilicates such as saponite and serpentine, magnetite, Fe-sulfide, and carbonates[1]. Chondrules and CAIs are very rare or absent, though isolated olivine and pyroxene grains inferred to be fragments of chondrules are observed[22–24]. It is not clear if the CI chondrites ever contained chondrules and CAIs and are essentially all matrix component, or if the chondrules and CAIs were consumed and their primary chondrite textures destroyed during extensive aqueous alteration[25]. It should be noted that Frank et al. [26] described a CAI with $^{16}O$-rich isotopic ratios in the Ivuna CI chondrite.

The Hayabusa2 spacecraft returned samples of ~5.4 g from C-type asteroid (162173) Ryugu[27]. The "stone" team, which is one of the six initial analysis teams, received 17 stone samples from the ISAS curation facility and conducted analyses for elucidation of early evolution of asteroid Ryugu[28]. The Ryugu samples mineralogically and chemically resemble CI chondrites[28–31]. Remote sensing observations by the Hayabusa2 spacecraft suggested that asteroid Ryugu formed by reaccumulation of rubble ejected by impact from a larger asteroid[32,33]. It was suggested that the Ryugu original parent body formed beyond the $H_2O$ and $CO_2$ snow lines in the solar nebula at 1.8–2.9 Myr after CAI formation[28], which is as early as formation of chondrules from major types of carbonaceous chondrites such as CM, CO, and CV at 2.2–2.7 Myr after CAI formation[3]. In addition, small chondrule-like objects and CAIs (<30 μm) were found in some Ryugu stone samples[28].

In this study, oxygen isotope analysis and further mineralogical and compositional analysis are performed on the chondrule-like objects and CAIs. This is the first detailed report of the chondrule-like objects and CAIs returned from known asteroid. The chondrule-like objects and CAIs are observed with field emission scanning electron microscope (FE-SEM) and analyzed for elemental compositions with field emission electron probe microanalyzer (FE-EPMA) and oxygen three-isotope ratios with secondary ion mass spectrometer (SIMS). A focused ion beam (FIB) section is taken out from one chondrule-like object and observed with field emission transmission electron microscope (FE-TEM). Our studies indicate that the chondrule-like objects and CAIs in the Ryugu samples have similarities and differences with chondrules and CAIs in chondrites. Here, we discuss the significance of the presence of chondrule-like objects and CAIs in asteroid Ryugu and their origins.

## Results

### Occurrence of chondrule-like objects and CAIs in the Ryugu samples

A small number of chondrule-like objects and CAIs are found by elemental mapping using FE-EPMA and FE-SEM observation of 42 polished sections from the 13 Ryugu samples (52.6 mm² in total). Chondrules and CAIs with sizes of ~100 μm–1 cm, which are typical for chondrites[1], are not observed[28]. The chondrule-like objects and CAIs analyzed for oxygen isotopes occur along with isolated olivine, pyroxene and spinel grains in the polished sections of C0040-02 and C0076-10 and in less-altered clasts (clast 1 and 2) in the polished section C0002-P5[28]. Fractions of the surface areas of all chondrule-like objects and CAIs observed in the Ryugu polished sections including those reported in Nakamura et al.[28] are estimated as ~15 ppm and 20 ppm, respectively, which are much smaller than those in carbonaceous chondrites[1].

### Mineralogy and chemistry of the chondrule-like objects in the Ryugu samples

Chondrule-like objects found in the Ryugu samples have rounded-to-spherical shapes with diameters of 10–20 μm (Fig. 1a–c), which are as small as chondrule-like Wild 2 particles[13]. Although remote sensing observations found mm-sized inclusions similar to chondrules on the surface of asteroid Ryugu[34], sizes of the chondrule-like objects that we found are much smaller. The chondrule-like objects analyzed for oxygen isotopes consist mainly of olivine with Mg# of ~99. Fe-Ni metal and sulfide are present in two of them. One object with no opaque minerals contains Al- and Ti-free diopside ($En_{56.0}Wo_{43.7}$; Supplementary Table 1). The three chondrule-like objects do not contain glass or glass-altered phase and are not surrounded by fine- or coarse-grained rim, unlike chondrules in chondrites[1]. In C0002-P5-C1-Chd, one out of three EPMA spots on Mg-rich olivine show a MnO/FeO ratio (wt%) exceeding 1 (Supplementary Table 1), which is characteristic for low-iron, manganese-enriched (LIME) olivine[35]. TEM analysis of the FIB section from C0040-02-Chd shows sub-μm-sized mixture of diopside and olivine with straight grain boundaries and well-developed 120° triple junctions (Fig. 2), which is evidence of annealing[36]. The sub-μm-sized olivine grains are LIME olivine (Supplementary Table 1). The 120° triple junctions are observed in olivine cores in chondrules and result of epitaxial growth of olivine during chondrule formation[37].

### Mineralogy and chemistry of the CAIs in the Ryugu samples

The two CAIs analyzed are ~30 μm in size (Fig. 1d, e), which are as small as CAI-like Wild 2 particles[12]. The two CAIs consist of spinel and hibonite along with tiny perovskite inclusions (detected by energy-dispersive X-ray spectrometry of FE-EPMA). Phyllosilicates with low totals of 69–93 wt% occur around the two CAIs and interstitial region of spinel grains in C0040-02-CAI and are free from opaque minerals such as Fe-sulfide and magnetite, unlike phyllosilicates of the surrounding Ryugu matrix (Fig. 1d, e). Phyllosilicates of the two CAIs have $Al_2O_3$ concentrations of 3.2–21.7 wt% (Supplementary Table 1), which are higher than those in the Ryugu matrix phyllosilicates (2.3 wt%)[28] and as high as those in phyllosilicates of altered CAIs in a CM carbonaceous chondrite (4.8–12.4 wt%)[38].

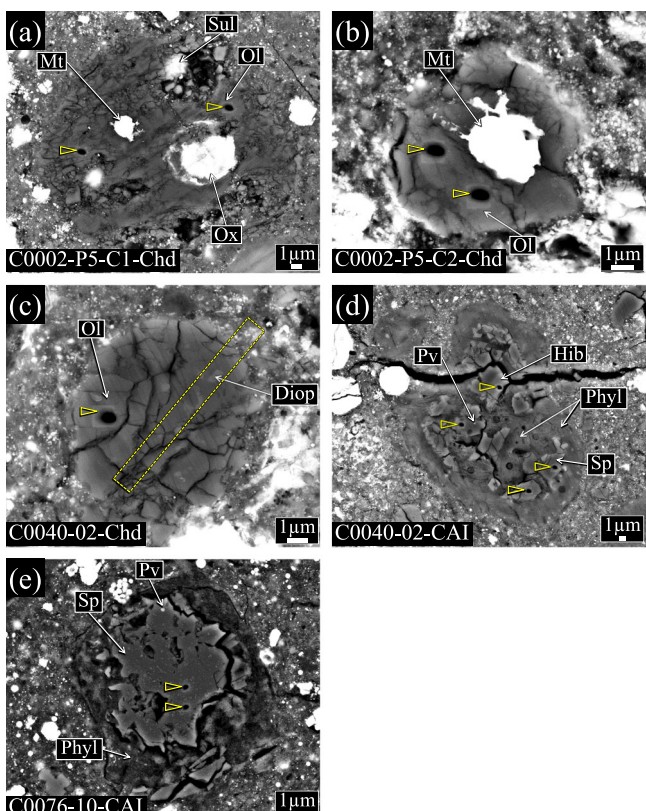

**Fig. 1 | Backscattered electron (BSE) images of three chondrule-like objects and two CAIs in the Ryugu samples analyzed for oxygen isotopes. a** C0002-P5-C1-Chd, **b** C0002-P5-C2-Chd, **c** C0040-02-Chd, **d** C0040-02-CAI, and **e** C0076-10-CAI. SIMS analysis spots are shown by the vertex of an open triangle. The rectangle area drown by the dashed line in panel (**c**) corresponds to the region extracted by the FIB sectioning. Ol, olivine; Mt, Fe-Ni metal; Sul, Fe-sulfide; Ox, oxide; Diop, diopside; Sp, spinel; Hib, hibonite; Pv, perovskite; Phyl, phyllosilicates.

## Oxygen-isotope ratios

We made a total of 11 spot analyses in the 3 chondrule-like objects and 2 CAIs. In each object, 1 to 4 spot analyses were made. A summary of the 11 spot analyses is shown in Table 1; a more complete information is given in Supplementary Table 2. The oxygen-isotope ratios show a bimodal distribution at peaks of ~ −43‰ and ~0‰ in $\delta^{18}O$ along the Carbonaceous Chondrite Anhydrous Mineral (CCAM) and the PCM lines[16,39] (Fig. 3), which is consistent with oxygen-isotope data of isolated olivine and pyroxene from the Ryugu samples[30,40,41]. The individual objects are isotopically uniform with the uncertainty of our measurements (see Supplementary Figs. 1–5). Two out of the three chondrule-like objects are $^{16}O$-rich with average $\Delta^{17}O$ values of −23.0 ± 6.0‰ (2σ; C0002-P5-C2-Chd) and −22.9 ± 5.2‰ (C0040-02-Chd; single spot); and the latter contains LIME olivine. The third object that contains LIME olivine is $^{16}O$-poor with average $\Delta^{17}O$ value of −3.4 ± 6.0‰ (C0002-P5-C1-Chd). The two CAIs are $^{16}O$-rich with average $\Delta^{17}O$ values of −22.5 ± 2.5‰ (C0040-02-CAI) and −24.2 ± 3.6‰ (C0076-10-CAI).

## Discussion

The three chondrule-like objects in the Ryugu samples are rounded-to-spherical objects dominated by olivine, which is characteristic for chondrules in chondrites[1]. One out of the three chondrule-like objects (C0002-P5-C1-Chd) has Mg# of 98.6, which is within the Mg# range of type I chondrules. The object has $^{16}O$-poor isotopic ratios with $\Delta^{17}O$ of −3.4 ± 6.0‰ (Fig. 3; Table 1), which is within the $\Delta^{17}O$ range (~ −5‰ to −2‰) of type I chondrules from carbonaceous chondrites[5] though with large uncertainty. Other two chondrule-like objects (C0002-P5-C2-Chd

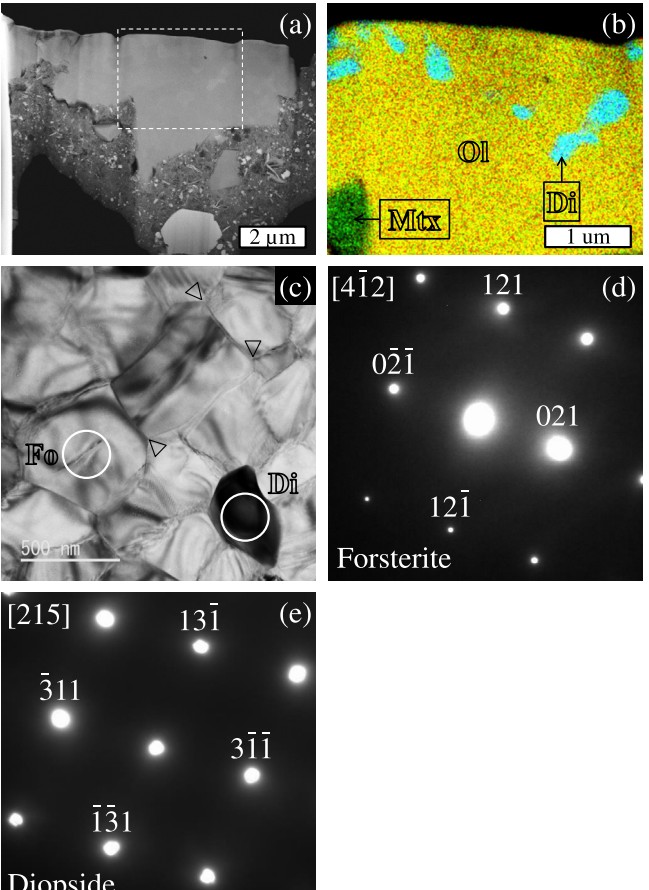

**Fig. 2 | Sub-micron structures of C0040-02-Chd. a** A high-angle annular dark-field (HAADF)-STEM image of the FIB section from C0040-02-Chd, **b** a combined elemental map in Mg (red), Si (green), and Ca (blue) X-rays of a rectangle area drawn with the dashed line in panel (**a**), **c** a bright field (BF)-TEM image of olivine and diopside in an area of C0040-02-Chd, and **d, e** selected-area electron diffraction (SAED) patterns from forsterite along the [4$\bar{1}$2] zone axis and diopside along the [215] zone axis. Abbreviations in panel (**b**): Ol, olivine; Di, diopside; Mtx, matrix. The 120° triple junctions are indicated by the vertex of an open triangle in panel (**c**). Circled areas in panel (**c**) represent analysis spots of electron diffraction on forsterite (Fo) and diopside (Di).

and C0040-02-Chd) are dominated by Mg-rich olivine and have $^{16}O$-rich isotopic ratios with $\Delta^{17}O$ of ~ −23‰ (Fig. 3; Table 1), which is within the $\Delta^{17}O$ range of CAIs and AOAs[6]. One of them (C0040-02-Chd) contains sub-μm-sized diopside and LIME olivine grains and shows an annealed texture with 120° triple junctions (Fig. 2), which are characteristic for AOAs[36,42,43].

Olivine in AOAs is depleted in refractory elements such as Ca, Al, and Ti compared with that in type I chondrules[44]. Relict olivine as $^{16}O$-rich as AOAs is also depleted in refractory elements compared with coexisting $^{16}O$-poor olivine in chondrules[20]. These trends are evident in Fig. 4 where CaO and $Cr_2O_3$ concentrations in olivine from AOAs and type I chondrules are compared. The olivine data plotted in Fig. 4 are only from type ≤3.0 chondrites (including aqueously-altered ones), because the original $Cr_2O_3$ concentrations in olivine are undisturbed only in type ≤3.0 chondrites[45,46]. Calcium and Cr are minor in olivine as indicated by the low concentrations of CaO and $Cr_2O_3$ in type I chondrule-olivine. This is because Ca is incompatible with olivine[47] and Cr is originally minor in chondrule precursor dust[5]. The AOA-olivine shows even lower concentrations of CaO and $Cr_2O_3$, which is explained by olivine condensation from a residual gas depleted in the refractory elements after condensation of refractory-rich minerals[48,49] followed by isolation from the gas before condensation of Cr. The CaO and

**Table 1 | Oxygen-isotope ratios of chondrule-like objects and CAIs in the Ryugu samples[a]**

| Sample name | Spot# | δ18O | 2SD (‰) | δ17O | 2SD (‰) | Δ17O | 2SD (‰) | Target[b] |
|---|---|---|---|---|---|---|---|---|
| C0002-P5-C1-Chd | 1 | 2.6 | 2.0 | −2.5 | 7.9 | −3.8 | 8.5 | Ol (Fo98.6) |
| | 2 | −1.4 | 2.0 | −3.7 | 7.9 | −3.0 | 8.5 | Ol |
| | Average | 0.6 | 3.9 | −3.1 | 5.6 | −3.4 | 6.0 | |
| C0002-P5-C2-Chd | 1 | −39.8 | 2.0 | −43.6 | 7.9 | −22.9 | 8.5 | Ol (Fo98.9) |
| | 2 | −47.5 | 2.0 | −47.8 | 7.9 | −23.1 | 8.5 | Ol |
| | Average | −43.6 | 7.7 | −45.7 | 5.6 | −23.0 | 6.0 | |
| C0040-02-Chd | 1 | −44.4 | 1.3 | −46.0 | 5.4 | −22.9 | 5.2 | Ol (Fo99.7) |
| C0040-02-CAI | 1 | −39.1 | 2.4 | −46.5 | 4.8 | −26.1 | 4.1 | Hib |
| | 2 | −43.1 | 2.4 | −42.7 | 4.8 | −20.2 | 4.1 | Sp |
| | 3 | −42.5 | 2.4 | −44.0 | 4.8 | −21.9 | 4.1 | Sp |
| | 4 | −43.1 | 2.4 | −44.2 | 4.8 | −21.8 | 4.1 | Sp |
| | Average | −42.0 | 1.9 | −44.3 | 2.4 | −22.5 | 2.5 | |
| C0076-10-CAI | 1 | −44.0 | 1.3 | −46.3 | 5.4 | −23.4 | 5.2 | Sp |
| | 2 | −40.3 | 1.3 | −46.0 | 5.4 | −25.1 | 5.2 | Sp |
| | Average | −42.1 | 3.7 | −46.1 | 3.8 | −24.2 | 3.6 | |

[a]The uncertainties associated with average values are twice the standard error of the mean (2SE).
[b]Average (or representative) chemical compositions are shown.

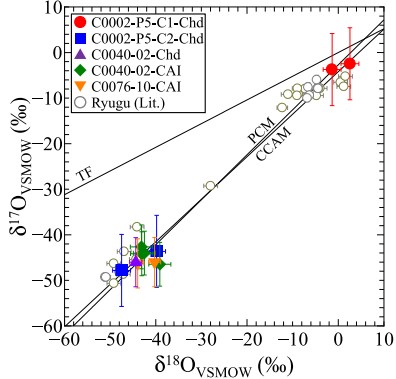

**Fig. 3 | Oxygen three-isotope ratios of three chondrule-like objects and two CAIs in the Ryugu samples.** TF, PCM, and CCAM represent the Terrestrial Fractionation line, the Primitive Chondrule Mineral line, and the Carbonaceous Chondrite Anhydrous Mineral line. Literature data of isolated olivine and pyroxene and AOA-like porous objects in the Ryugu samples are plotted for comparison[30,41].

$Cr_2O_3$ concentrations in olivine in the two [16]O-rich chondrule-like objects plot in the range of the AOA-olivine, while those in olivine in the [16]O-poor one plot in the range of type I chondrule-olivine (Fig. 4). Thus, the [16]O-poor chondrule-like object shares characteristics with type I chondrules in carbonaceous chondrites, and two [16]O-rich ones share characteristics with AOAs. Likewise, the CaO and $Cr_2O_3$ concentrations in the [16]O-rich isolated olivine grains from CI chondrites and the Ryugu samples plot in the range of the AOA-olivine, while those in the [16]O-poor ones plot in the range of type I chondrule-olivine[22–24,30,41]. It is worth mentioning that CaO and $Cr_2O_3$ concentrations in olivine in the anomalously [16]O-rich chondrule from a CH chondrite[50] plot in the range of the AOA-olivine (Fig. 4), suggesting a genetic link to AOAs.

AOAs are characterized by irregular shapes, numerous pores, and refractory minerals including anorthite, Al-diopside, and spinel, besides Mg-rich olivine, though some AOAs are compact coarse-grained objects containing subhedral-to-euhedral diopside grains[10,43,51]. The two [16]O-rich chondrule-like objects are rounded and free from pores and refractory minerals (Fig. 1b, c). It is less likely that the two objects are AOA fragments with no pores and refractory minerals, given that [16]O-rich isolated olivine in the Ryugu samples and CI chondrites which are suggested to

be AOA fragments have angular shapes[24,40]. One of the [16]O-rich chondrule-like objects contains a rounded Fe-Ni metal grain (Fig. 1b) that solidified from a molten metal droplet and may have experienced a melting event. Thus, the two [16]O-rich chondrule-like objects are likely to have been originally AOAs (or fragments) and melted (and annealed) by a heating event in the [16]O-rich environment possibly near the Sun.

Chondrules are products of multiple heating events[5,19]. Remnants of the early generations of chondrules are observed as relict grains in chondrules[5,19–21,52], of which characteristics are similar to those of the three chondrule-like objects in the Ryugu samples; e.g., Mg-rich olivine-dominated mineralogy and [16]O-rich isotope signatures. Here we discuss the possibility that the three chondrule-like objects are early generations of chondrules.

Chondrules in chondrites are diverse in texture, but they commonly contain glassy mesostasis, except for cryptocrystalline chondrules[1]. Differently, the three chondrule-like objects are free from glass (or glass-altered phase) and are dominated by Mg-rich olivine along with Fe-Ni metal and sulfide (Fig. 1a–c), which are similar to what has been proposed as early generations of chondrules in Libourel and Krot[52]. Especially, one of the three chondrule-like objects show an annealed texture (Fig. 2), like early generations of chondrules proposed in Libourel and Krot[52]. It is therefore suggested that the three chondrule-like objects are early generations of chondrules. The early generations of chondrules suggested in Libourel and Krot[52] are products from differentiated planetesimals, but which cannot provide objects with variable oxygen-isotope ratios of [16]O-rich and -poor observed in the present study (see also Marrocchi et al.[37]). Instead, the diverse oxygen isotope compositions of the three chondrules is consistent with nebular products as suggested in Whattham et al.[53]. Agglomeratic olivine (AO) chondrules are also one of what have been proposed as earlier generations of chondrules[54], but which are different from the chondrule-like objects found in this study. The AO chondrules consist of olivine and pyroxene grains with variable Mg# and various sizes that are lightly sintered. Chondrules in chondrites contain relict olivine, which are generally more [16]O-rich than coexisting mineral phases[5]. Such [16]O-rich relict olivine is likely to be a remnant of earlier generations of chondrules or fragments of AOAs[5,20,21]. The two [16]O-rich chondrule-like objects may be early generations of chondrules that escaped from incorporation into [16]O-poor chondrule precursor dust. Recently, Marrocchi et al.[55] reported that smaller chondrules tend to be more [16]O-rich than larger ones in CR chondrites and suggested

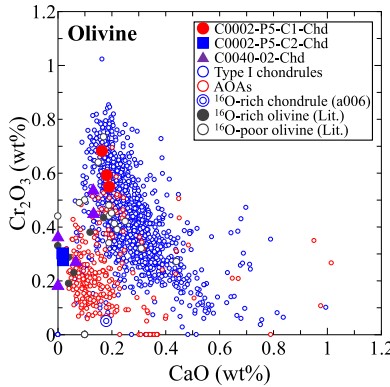

**Fig. 4 | Comparison of concentrations between $Cr_2O_3$ and CaO in olivine from the three chondrule-like objects in the Ryugu samples.** Concentrations of $Cr_2O_3$ and CaO in olivine from C0040-02-Chd are only from TEM-EDS data, as the EPMA data is mixture of olivine and diopside. Olivine data of type I chondrules and AOAs, which are plotted for comparison, are from type ≤3.0 chondrites[10,16,17,21,36,37,42,43,45,46,49,51,55,56,71,80–86]. Concentrations of $Cr_2O_3$ and CaO in olivine in the [16]O-rich chondrule (a006) from a CH chondrite[50] and those in [16]O-rich and -poor olivine from the Ryugu samples and CI chondrites[22–24,30,41] are plotted for comparison.

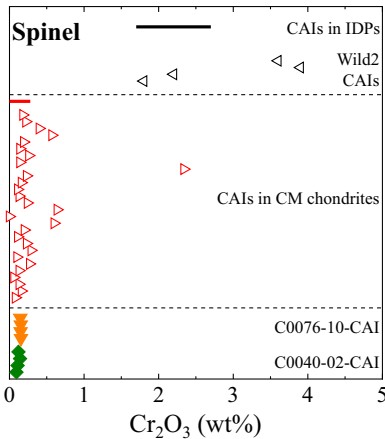

**Fig. 5 | Concentrations of $Cr_2O_3$ in spinel from the two Ryugu CAIs.** Spinel data in CAIs from CM chondrites[38,87–92], CAI-like Wild 2 particles[62], and CAI-like IDP[14] are shown for comparison. The red and black bars represent the $Cr_2O_3$ ranges in spinel in CAIs in a CM chondrite[92] and in CAI-like IDP (Spray)[14].

that the relatively [16]O-rich small chondrules escaped from incorporation into [16]O-poor CI-like dust, which is consistent with our interpretation described above.

If the three chondrule-like objects in the Ryugu samples are early generations of chondrules, the two distinct oxygen-isotope ratios of [16]O-rich and -poor (Fig. 3) are evidence for the argument that [16]O-rich (~ −23‰ in $\Delta^{17}O$) and [16]O-poor (~0‰) isotope reservoirs existed in the early stage of the chondrule formation[5,37,56,57]. While [16]O-poor chondrules are commonly observed in chondrites[5], [16]O-rich chondrules are very rare[50]. Only [16]O-rich relict grains are observed as minor constituents in chondrules[13,20,21]. A possible explanation is that the [16]O-rich chondrules were incorporated into [16]O-poor chondrule precursor dust and reheated, as described above. Even if the [16]O-rich chondrules escaped from the recycling events, they should have been incorporated into early-formed planetesimals such as parent bodies of differentiated meteorites (0.5–1.9 Myr after CAIs)[58] and destroyed during the differentiation processes. The reason for the presence of [16]O-rich (and -poor) chondrule-like objects in the Ryugu samples is discussed in the final section.

The two Ryugu CAIs consist of spinel, hibonite, and perovskite and have [16]O-rich isotopic ratios (Figs. 1d, e and 3), which are characteristic for CAIs in chondrites[6]. The refractory minerals of the CAIs are embedded in and/or surrounded by Al-rich phyllosilicates. Unlike phyllosilicates in the Ryugu matrix, Al-rich phyllosilicates are free from opaque minerals such as magnetite and sulfide but contain certain amounts of $SO_3$ (0.8–5.6 wt%; Supplementary Table 1). Tiny sulfide grains that are unrecognizable under FE-SEM may be present in Al-rich phyllosilicates. It is likely that Al-rich phyllosilicates surrounding the two Ryugu CAIs are originally an Al-rich mineral phase susceptible to aqueous alteration such as melilite or anorthite[6,38]. In this case, the depletion in Ca in Al-rich phyllosilicates (<0.6 wt%; Supplementary Table 1) is attributed to mobilization of this element to form calcite during aqueous alteration[59].

Spinel-hibonite inclusions accompanied by altered phases, like C0040-02-CAI, and spinel inclusions surrounded by altered phases, like C0076-10-CAI, are observed in CM chondrites[38,60]. However, the two Ryugu CAIs are smaller than CAIs in CM chondrites and as small as CAI-like Wild 2 particles[12]. The cometary CAIs are younger than the CM-CAIs, which are as old as the oldest CAIs[8,60,61]. In addition, the cometary CAIs contain relatively high concentrations of $Cr_2O_3$ compared with CAIs in chondrites[62]. It is therefore suggested that the cometary

CAIs experienced remelting events with addition of less refractory elements after initial formation[62]. Spinel is the only common mineral between the two Ryugu CAIs (perovskite is too tiny to analyze elemental compositions precisely) and occurs in the CM-CAIs and cometary CAIs. Here we discuss whether the two Ryugu CAIs resemble CM-CAIs or cometary CAIs based on the $Cr_2O_3$ concentrations (Fig. 5), which may facilitate estimation of timing of the two Ryugu CAI formation. Spinel in the CM-CAIs contain $Cr_2O_3$ mostly less than 0.6 wt%, while that in CAI-like Wild 2 particles and CAI-like IDP contains more $Cr_2O_3$ than 1.7 wt%. Matzel et al.[61] suggested that the CAI-like Wild 2 particle, Coki, is classified into type C CAIs, which experienced remelting events[63]. The high concentrations of $Cr_2O_3$ in the cometary CAIs and type C CAIs are explained by addition of Cr from Cr-bearing gas or dust during the remelting events in the chondrule-forming regions[62,64]. Based on the $^{26}Al$-$^{26}Mg$ chronometry, CAI-like Wild 2 particles do not show $^{26}Mg$ excess and are younger (few Myr or more)[61,65] than CM-CAIs[60], which reflects the relatively late remelting events. Hibonite-rich CAIs, one of the CM-CAI groups, show no resolvable $^{26}Mg$ excess due to in-situ $^{26}Al$ decay and appear to be young, but which have formed before injection or widespread distribution of $^{26}Al$ in the solar nebula[66]. The two CAI-like Wild 2 particles are mineralogically different from the hibonite-rich CAIs and are most likely young objects. Spinel in the two Ryugu CAIs contain $Cr_2O_3$ less than 0.2 wt% (Fig. 5). It is possible that the two Ryugu CAIs escaped from remelting events that supplied Cr. If this is the case, the two Ryugu CAIs may possibly be as old as the CM-CAIs.

We found three chondrule-like objects that are likely to be early generations of chondrules (two of them have affinities to AOAs) and two CAIs that may possibly be as old as the oldest CAIs based on the mineralogy, chemistry, and oxygen-isotope ratios. Additional important observations in the present study are the small sizes (<30 μm) and rarity (~15 ppm and 20 ppm) of chondrule-like objects and CAIs in the Ryugu samples. Isolated olivine, pyroxene, and spinel grains that are likely to be fragments of chondrules and CAIs and AOA-like porous olivine in the Ryugu samples are also small (<30 μm)[28,40,41]. The Ryugu original parent body formed beyond the $H_2O$ and $CO_2$ snow lines in the solar nebula (>3–4 au from the Sun)[28], while CAIs and AOAs formed near the Sun[6] or planet-forming regions (~1 au)[7]. Radial transport of CAIs and AOAs from the inner regions to the region where the Ryugu original parent body formed is required. The two [16]O-rich chondrule-like objects formed near the Sun may have been transported along with CAIs. Likewise, it has been suggested from the observations of chondrule-like and CAI-like Wild 2 particles that chondrules and CAIs

were transported from the inner regions to the Kuiper belt (-30–50 au) in the solar nebula[12,13,67]. Chondrule-like fragments in the giant cluster IDP, which is cometary in origin, are as small as those from comet Wild 2[15]. Given the smaller sizes of the cometary chondrules and CAIs than those in chondrites, radial transport favoring smaller objects to farther locations may have occurred in the solar nebula; e.g., a combination of advection and turbulent diffusion[68]. If this is the case, the occurrence of chondrule-like objects and CAIs in the Ryugu samples as small as those in the Wild 2 particles suggests that the Ryugu parent body formed at farther location than any other chondrite parent bodies and acquired $^{16}O$-rich and -poor chondrule-like objects and CAIs transported from the inner solar nebula.

Chondrules in different chondrite groups have distinct chemical, isotopic, and physical properties, which suggests chondrule formation in local disk regions and subsequent accretion to their respective parent bodies without significant inward/outward migration[5,69–71], though with a limited number of ordinary chondrite chondrules being observed in carbonaceous chondrites[72]. It is considered from the rarity of chondrules (and chondrule-like objects) in the Ryugu samples that the Ryugu original parent body formed in a region scarce in chondrules. Instead, small chondrules (and chondrule-like objects) and their fragments may have been transported from the inner solar nebula and accreted along with CAIs onto the Ryugu original parent body. Since the formation age of the Ryugu original parent body (1.8–2.9 Myr after CAI formation)[28] is as early as those of major types of carbonaceous chondrite chondrules such as CM, CO, and CV (2.2–2.7 Myr after CAI formation)[3], chondrules typically observed in chondrites (100 μm–1 mm)[1] should have presented in the inner regions of the solar nebula when forming the Ryugu original parent body. Considering radial transport favoring smaller objects to the formation location of the Ryugu original parent body, fragments of the relatively large chondrules may have been provided and observed as isolated olivine and pyroxene grains in the Ryugu samples. Recently, Morin et al.[24] analyzed oxygen-isotope ratios of isolated olivine and low-Ca pyroxene grains in CI chondrites. Although they suggested that $^{16}O$-poor grains are fragments of chondrules formed in the CI chondrite formation regions, the reason for the limited size range of the most isolated grains (<30 μm) compared with that for other carbonaceous chondrites (up to -200 μm in diameter)[73] is unclear.

CAIs in the Ryugu samples are much less abundant (-20 ppm) than those in the Wild 2 particles (-0.5%)[62], suggesting destruction of the CAIs and chondrules (and chondrule-like objects) in the Ryugu original parent body during the extensive aqueous alteration. Since the chondrule-like objects and CAIs occur along with isolated anhydrous grains in less-altered clasts and samples, these objects may have survived in less-altered regions in the Ryugu parent body but have not been incorporated into the Ryugu parent body after the aqueous alteration or asteroid Ryugu.

## Methods
### Sample preparation
Polished sections were prepared from the Ryugu samples C0002, C0040, and C0076 based on the methods dedicated to the Ryugu samples[74]. C0002-P5 (C0002-Plate5 in Nakamura et al.[28]) means 5th plate of six plates from C0002. C0040-02 and C0076-10 mean 2nd polished section from C0040 and 10th polished section from C0076. The polished sections were coated with carbon (20–30 nm in thickness). C0002-P5 was loaded in the 3-hole disk[75], and other two polished sections were loaded in the 7-hole disk[75] for electron microscopy and oxygen-isotope analysis with SIMS. The chondrule-like objects and CAIs analyzed for oxygen isotopes are located outside of the 500 μm and 1 mm radius of the center of holes for 7-hole and 3-hole disks, which allow accurate SIMS analysis within ±0.5‰ in $\delta^{18}O$ with -10 μm primary

beam (-2 nA)[75]. But the analytical uncertainty of the oxygen-isotope analysis of the chondrule-like objects and CAIs is more than ±1‰ in $\delta^{18}O$ as described later, so that the instrumental mass bias is insignificant.

### Electron microscopy
Chondrule-like objects and CAIs in the Ryugu samples were examined using a FE-SEM (JEOL JSM-7001F) at Tohoku University, and BSE images were obtained. Elemental compositions of the chondrule-like objects and CAIs were measured using a FE-EPMA (JEOL JXA-8530F) equipped with wavelength-dispersive X-ray spectrometers (WDSs) at University of Tokyo. WDS quantitative chemical analyses of olivine in the chondrule-like objects and spinel and hibonite in the CAIs were performed at 12 kV accelerating voltage and 30 nA beam current with a focused beam. For analyses of phyllosilicates of the CAIs, 15 kV accelerating voltage and 12 nA beam current with a defocused beam of 1 μm were applied. Natural and synthetic standards were chosen based on the compositions of the minerals being analyzed[28].

A FIB section from C0040-02 was extracted using a FIB-SEM (Thermo Fischer Scientific Versa 3D) at Tohoku University for TEM observation. The region of interest was coated by platinum deposition to prevent damage during FIB processing. Then, it was cut out as a thick plate (-1 μm in thickness) and mounted on copper grids and thinned to 100–200 nm using a $Ga^+$ ion beam at 30 kV and 0.1–2.5 nA. The damaged layers formed on the thin sections during the thinning were removed using a $Ga^+$ ion beam at 5 kV and 16–48 pA.

The thin section was observed with a FE-TEM (JEOL JEM-2100F) operating at 200 kV and equipped with an energy-dispersive X-ray spectrometer (EDS) at Tohoku University. TEM images were recorded using a charge-coupled device (CCD) and then processed by the Gatan Digital Micrograph software package. Crystal structures were identified based on analysis of SAED patterns. We also acquired STEM images. X-ray maps and quantitative EDS data were obtained using JEOL JED-2300 EDS detectors and JEOL analysis station software package. Quantifications of EDS spectra were carried out using the Cliff-Lorimer thin film approximation using theoretical k-factors.

### Oxygen-isotope analysis
Before the oxygen-isotope analysis of chondrule-like objects and CAIs in the Ryugu samples, FIB markings were employed at selected locations of each object, which were identified by the $^{16}O^-$ secondary ion imaging[76,77]. Accurate aiming using FIB marking and $^{16}O^-$ ion imaging avoids significant beam overlap with adjacent mineral phases, so that accurate oxygen-isotope ratios are obtained. A FIB-SEM (Thermo Fischer Scientific Helios NanoLab 600i) equipped with a gallium ion source at Tohoku University was used to remove surface carbon coating from the chondrule-like objects and CAIs. A 30 kV focused $Ga^+$ ion beam set to 7 pA was rastered within a 1 μm × 1 μm square on the sample surface for 30 s, so that only the surface coating was removed without significant milling of underlying mineral. This 1 μm square region was later identified by secondary $^{16}O^-$ ion imaging in SIMS before oxygen-isotope analysis.

Oxygen-isotope ratios of three chondrule-like objects and two CAIs in the Ryugu samples were analyzed with the CAMECA IMS 1280 at the University of Wisconsin-Madison. The analytical conditions and measurement procedures were similar to those in Zhang et al.[77]. A focused $Cs^+$ primary beam was set to -0.8 μm × 0.5 μm and intensity of -0.3 pA. The secondary $^{16}O^-$, $^{17}O^-$, and $^{18}O^-$ ions were detected simultaneously by a Faraday Cup ($^{16}O^-$) with $10^{12}$ ohm feedback resistor and electron multipliers ($^{17}O^-$, $^{18}O^-$) on the multi-collection system. Intensities of $^{16}O^-$ were -2–3 × $10^5$ cps. The contribution of the tailing of $^{16}O^1H^-$ interference to $^{17}O^-$ signal was corrected by the method described in Heck et al.[78], though the contribution was negligibly small (≤0.5‰). One to four analyses were performed for each object, bracketed by six analyses (three analyses before and after the unknown sample analyses) on the San

Carlos olivine (SC-Ol) grains mounted in the same multiple-hole disks. The external reproducibility of the running standards was 1.3–2.4‰ for $\delta^{18}O$, 4.8–7.9‰ for $\delta^{17}O$, and 4.1–8.5‰ for $\Delta^{17}O$ (2 SD; standard deviation), which were assigned as analytical uncertainties of unknown samples; see Kita et al. [17] for detailed explanations. We analyzed olivine ($Fo_{100}$), spinel, and hibonite standards[17,79] in the same session for correction of instrumental bias of olivine, spinel, and hibonite. Instrumental biases estimated from above mineral standards (matrix effect) are within a few ‰ in $\delta^{18}O$ (Supplementary Table 3). After SIMS analyses, all SIMS pits were inspected using a FE-SEM to confirm the analyzed positions (Fig. 1).

## Data availability
The elemental and oxygen isotope data generated in this study are provided in the Supplementary Information Data file.

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

## Acknowledgements

We are grateful to K. Sato for help with FIB marking, T. Miyazaki for help with TEM observation, and M. J. Spicuzza and K. Kitajima for SIMS support. This work was supported by JSPS KAKENHI grant numbers 18H01263 (D.N.) and JP20H00188 and 21H00159 (T.N.). WiscSIMS is partly supported by NSF (EAR 2004618).

## Author contributions

Study was conceived and designed by D.N. and T. Nakamura. Sample preparation by D.N. Scanning electron microscopy by T. Nakamura, D.N., and T. Morita. Electron microprobe analysis by T. Nakamura, T. Mikouchi, and H. Yoshida. Oxygen isotope analysis by M.Z., N.T.K., T. Morita, and D.N. Transmission electron microscopy by Y.E., T. Morita, D.N. D.N. interpreted the data and wrote the paper with input from T. Nakamura, N.T.K., M.Z., T. Mikouchi, H. Yurimoto, and S. Tachibana. M.K., K.A., E.K., T.Y., M.N., A.N., A.M., K.Y., M.A., T.O., T.U., M.Y., T.S., S. Tanaka, S.N., F.T., T. Noguchi, H. Yabuta, H.N., R.O., K.S., S.W., and Y. T. assisted with the analyses.

## Competing interests

The authors declare no competing interests.
