## [Peer Review File · Nature Communications]

Chondrule-like objects and Ca-Al-rich inclusions in asteroid Ryugu are the earliest Solar System objectsREVIEWER COMMENTS

Reviewer #1 (Remarks to the Author):

This paper reports the presence of chondrule-like objects and CAIs in the Ryugu samples brought back to Earth by the Japanese space mission Hayabusa2. The authors determined the chemical and oxygen isotopic compositions of three chondrule-like objects and two CAIs. They used the data for discussing the origin of chondrules and the transport into the accretion disk. Although the data are interesting, they are not really new as similar results were already published for both CI and Ryugu samples. I do not really understand why different working groups of the Hayabusa2 space mission try to publish similar data in different papers (this sounds like salami science to me and one single paper should be enough). In addition, I found that the interpretation is rather limited and not really put into perspective with the different (and some recent) models of CAI and chondrule formation.

1. Similar data have already been published for the CI chondrites Alais and Ivuna (Morin et al., 2022, GCA) and for Ryugu particles (Nakamura et al., 2022 Proc. Jpn. Acad., Ser.B 98). Although these papers are cited, the authors should report these data into the chemical and isotopic diagrams. In addition, the authors did not take into account the data reported for mechanically-extracted olivine grains from CI chondrites (Leshin et al., 1997, Piralla et al., 2020).

2. In a general manner, the citations are very biased and mainly cite the work of the Wisconsin/American groups on chondrules, whereas other data exist, notably on the chemistry of relict olivines (see the work by the Nancy group). For instance, it has been shown that relict olivine grains in chondrules are Ca-Al-Ti depleted whereas host olivines are not. This has been observed in both CC (Marrocchi et al., 2018, 2019, 2022), OC (Piralla et al., 2021) and R chondrites (Regnault et al., 2022). Why not comparing your data to this dataset for discussing your result?

3. Line 170. What do you mean by type < 3.0 chondrites? Types 2 and 1 that experienced aqueous alteration? If so, it does not sound like a good idea.

4. It has been recently shown that small chondrules in CR chondrites are ^{16}O -rich compared to larger chondrules (Marrocchi et al., 2022). In addition, a similar observation was reported long time ago by Bob Clayton for small chondrules in ordinary chondrites (Clayton et al., 1991). The authors should consider these studies for discussing their data. In addition, ^{16}O -poor relict olivine grains compared to hosts have also been reported in CC chondrules (Marrocchi et al., 2018), thus confirming that both ^{16}O -rich and ^{16}O -poor isotopic reservoirs could have existed in the early solar system.

5. Lines 211-216. This part shows a profound miss-conception of the SiO model proposed by Libourel, Dominik Hezel and the Nancy group. What appears unlikely is that SiO does not play a role in chondrule formation.

-First, we all agree that the Chaussidon et al. 2008 paper is wrong and there is no O-isotopic difference between host olivine and low-Ca pyroxene grains. It likely comes from a calibration problem of the matrix effect (no low-Ca pyroxene was available at this time in Nancy).

-The Marrocchi and Chaussidon 2015 paper is a model. As all models, it does not pretend to explain all the data. However, it shows that different dust/gas ratios can generate isotopic disequilibrium between olivine and low-Ca pyroxene (independently if it is observed or not in chondrules)

-Given that said, it does not exclude SiO as the ^{16}O -poor endmember. Several papers recently reported complex textures in olivines of porphyritic chondrules, especially epitaxial growth of olivines in the outer part of chondrules (Libourel and Portail, 2018; Libourel et al., 2022; Marrocchi et al., 2018, 2019; Jacquet et al., 2021, Piralla et al., 2021). These textures require the addition of Mg and SiO from the gas phase. If the authors think it can be achieved with water instead of Mg/SiO, they can propose an alternative.

-The Mg/SiO model does not predict that host olivine and low-Ca pyroxene grains to have different oxygen isotopic composition because both formed during gas-melt interactions. Water may have played a role in controlling the oxygen fugacity and part of the O-isotopic compositions but it appears extremely difficult to explain the textures without Mg/SiO (which also participate to the O isotopic budget).

-Claiming that Mg/SiO is unlikely is also difficult to reconcile with the presence of olivine and pyroxene (and even silica sometimes) in chondrules. How you will produce this mineralogical diversity without invoking SiO coming from the gas phase? If you only consider water, there is no reason you will crystallize olivine and pyroxene (and silica of course) in the same chondrule. In addition, low-Ca pyroxenes are currently observed in the outer part of chondrules, parallels to the surface. How can explain that without an increase of the silicon activity in chondrules?

-Another isotopic constraint that supports Mg/SiO is the large Si isotopic compositions observed in type I porphyritic chondrules, which is impossible to explain without SiO.

Hence, there are a lot of evidence of the role played by Mg/SiO during chondrule formation. Trying to dismiss this model with only one weak argument is not serious considering the large dataset available in literature (which you did not take into account).

6.Lines 284-286. There is a current debate today if chondrules accreted where they formed or not (Williams et al., 2020 vs. Schneider et al. 2020 for instance). If your data (now) support no transport, it would be great to explain this debate and the different model that have been proposed. This can be tested by plotting the D17O versus D18O (d18O-d17O). It has been recently shown that there is a difference between chondrules formed in the inner part and the outer part of the disk using this diagram (Piralla et al., 2021).

7.The discussion on why CAIs are observed in CI/Ryugu samples does not take into account the recent models where CAIs could have been formed further away from the Sun than previously thought, during the disk viscous spreading.

For instance, the dissipation of turbulence could sustain CAI-forming temperatures at distances of several tenths of an AU, especially at early times during disk build-up where the massive disk could retain heat more efficiently (Hueso and Guillot, 2005; Dullemond et al., 2006; Yang and Ciesla, 2012; Pignatale et al., 2018). In addition, the formation of CAIs/AOAs at larger heliocentric distances is now supported by recent geochemical measurements (V isotopes) suggesting that CAIs experienced low levels of irradiation that are difficult to reconcile with their formation at the disk inner edge (Bekaert et al., 2021). In addition, carbonaceous chondrites, including CIs, contain the most refractory inclusions of any chondrite class, even though they are thought to have accreted in the outer solar system at <200 K (Warren, 2011). This paradox can be mitigated if (i) refractory inclusions were entrained by the initial outward expansion of the disk (e.g., Pignatale et al. 2018) and/or (ii) preferentially trapped at the outer edge of the gap carved by growing Jupiter (e.g., Desch et al., 2018).

8.A CAI has also been reported in the CI Ivuna and should be mentioned (Frank et al., 2011, 2017).

Reviewer #2 (Remarks to the Author):

Title: Chondrule-like objects and CAIs in asteroid Ryugu: earlier generations of chondrules

Reviewer: Devin L. Schrader

Dear Authors,

The data is of exceptional quality, technically sound, and presented in detail. The results are exciting, will be of broad interest to the scientific community, and will advance the field by understanding the

formation/composition of these small chondrules, CAIs, and asteroid Ryugu. The conclusions are adequately supported.

I enjoyed reading this manuscript and I hope you find my review useful. I look forward to seeing the manuscript published.

Comments:

1. The manuscript needs to include and would benefit from an acknowledgement and comparison to agglomeratic olivine (AO) chondrules that we found in the Renazzo-like carbonaceous (CR) chondrites. We demonstrated that these AO chondrules were indistinguishable (size, texture, chemical, and O-isotopic composition) from the chondrule-like objects found in Comet Wild 2 (Schrader et al., 2018-GCA; <https://doi.org/10.1016/j.gca.2017.12.014>). The objects we found in CR chondrites have the same size and O-isotopic composition as those objects seen by these authors in Ryugu, and we argued for a similar origin from amoeboid olivine aggregates very early in the Solar system, and outward migration of these objects well.

Schrader D. L., Nagashima K., Waitukaitis S. R., Davidson J., McCoy T. J., Connolly Jr. H. C., and Lauretta D. S. (2018) The retention of dust in protoplanetary disks: Evidence from agglomeratic olivine chondrules from the outer Solar System. *Geochim. Cosmochim. Acta* 223, 405–421.

Relevant citation locations/areas to address the results of Schrader et al. (2018):

Lines 74; 117/118+; 166/167; 183 – 190; ‘Earlier Generation of Chondrules’ Section starting at Line 192; 221; 280 – 283. Could also add EPMA data from Schrader et al. (2018) on AO chondrules to Fig. 4 (CaO vs. Cr₂O₃ wt.%).

2. “...resembling what proposed as earlier generations of chondrules”

-Change to: resembling what has been proposed as earlier generations of chondrules?

3. I think Comet Wild 2 should have a space between Wild and 2. Wild2 is used throughout the current draft. See Zolensky et al. (2006-Science; doi: 10.1126/science.1135842).

4. Lines 88-89. “...which is as early as formation of major types of carbonaceous chondrite chondrules at 2.2 – 2.7 Myr after CAI formation...”

-Since this isn't the age range of chondrule formation for all types of carbonaceous chondrites (i.e., CR chondrules formed later), I recommend being specific about which meteorite types are meant here.

5. Update reference #54 from the LPSC abstract to the very recently published paper.

Liu et al. (2022) Incorporation of 16O-rich anhydrous silicates in the protolith of highly hydrated asteroid Ryugu, *Nature Astronomy*. <https://www.nature.com/articles/s41550-022-01762-4>

Reviewer #3 (Remarks to the Author):

The paper reports on the mineralogy, petrography, and O-isotope compositions of CAIs and igneous-like rounded objects in the unique Ryugu samples returned by the JAXA space mission. These are the first detailed descriptions of such objects in Ryugu samples: all previous studies described only isolated anhydrous silicates. The major conclusion of the paper that Ryugu samples contain small (< 30 μm) CAI- and chondrule-like objects that formed in the inner protoplanetary disk and were transported outward, to the accretion region of the Ryugu parent body compositionally similar to CI-chondrites. Contrary to other chondrite groups this region apparently avoided chondrule formation. The data are of high quality and the interpretation of the results obtained is reasonable, but not without caveat: chronology of objects cannot be uniquely inferred from the data obtained. It is a pity that not attempts were made for measuring Al-Mg systematics in hibonite of one of the CAIs described. It may be too difficult considering small grain sizes or it may be a goal of future studies.

These are still relatively minor issues and I am glad to recommend the paper for publication in Nature Communications after moderate revision. I hope that my comments listed below will help the authors to improve the paper.

Detailed comments:

1. The title does not exactly reflect the content of the paper. Although the CAIs in the Ryugu samples are possibly among the oldest refractory inclusions (the expectation is that these CAIs are ^{26}Al -poor), there is no chronological information in the paper. In addition, igneous CAIs cannot be considered to be earlier generations of chondrules: chondrules and igneous CAIs were melted in different disk regions characterized by different O-isotope compositions and ambient temperatures.
2. Line 56 – Please add U-corrected Pb-Pb ... and add a reference to Connelly et al. (2012) Science
3. AOA's are aggregates of solar nebular condensates. Since forsterite has lower condensation temperatures than most minerals in CAIs, AOA's represent lower temperature condensates than CAIs.
4. Oxygen-isotope (please use hyphen) & oxygen isotopic (no hyphen)
5. Line 65 – Chondrules are isotopically uniform only in unmetamorphosed (petrologic types ≤ 3.0). It may be better to define $\delta^{17}\text{O}$ and $\delta^{18}\text{O}$ and use them in the text instead of "oxygen-isotope ratios" which are too often used in lines 65-71.
6. Line 68 – It may be better to use PCM line instead of "slope-1 line". Note that PCM is used in line 148.
7. Line 75 - ... phyllosilicates
8. Line 76 - ... carbonates
9. Line 78 Frank et al. described polymineralic CAI in CI chondrite. It is uniformly ^{16}O -rich and quite refractory.
10. Line 90-91 – Are the Ryugu samples or CAIs and chondrules smaller than 30 μm ?
11. Line 92 - Not all elements analyzed are major.
12. Line 96 – Delete "As a result". You may say "Our studies indicate that ..."
13. Line 110 – You may need to mention that chondrule-like objects and CAIs were discovered in C samples which are less altered than A samples.
14. Line 117 - I am not sure if it is allowed to refer to a submitted paper.
15. Line 134 – it may be better to use "inclusions" instead of "particles".
16. Please use "phyllosilicates" in the text.
17. Line 183 – Some AOA's are compact coarse-grained objects containing subhedral-to-euhedral diopside grains; they appear to have experienced melting.
18. Line 193 – I don't know chondrules having ^{16}O -rich isotopic compositions except a single chondrule described by Kabayashi et al.
19. Line 199 - ... chondrule-like ...
20. Note that igneous CAIs and AOA's (some AOA's have been melted) experienced annealing and contain no glass, but we do not call them early generations of chondrules. Same comment for line 211.
21. Subsequent TEM studies showed that objects described by Libourel and Krot as "fragments of differentiated planetesimals" in fact contain glass along "dry contacts", i.e. they were incorrectly interpreted. It may be unnecessary to discuss in detail their paper.
22. The ^{16}O -poor counterpart is dust dominated the chondrule-forming regions characterized by high dust/gas ratios. The SiO gas results from evaporation of such disk regions.
23. Line 217-219. The co-existence of ^{16}O -rich and ^{16}O -depleted gaseous reservoirs during melting of refractory dust has been inferred in several papers (e.g., Kawasaki et al., 2018 and references therein).
24. Line 244 It is not clear why cometary CAIs are younger than CM CAIs. Is it based on Al-Mg isotope systematics? Note that CM PLACs have no radiogenic ^{26}Mg excess whereas SHIBs have ^{26}Mg excess corresponding to the canonical $^{26}\text{Al}/^{27}\text{Al}$ ratio.
25. Line 245-246. Do all cometary CAIs measured for O-isotope composition have uniform ^{16}O -rich compositions (it seems like this is not the case – see Fig. 13 in Joswiak et al. 2017, MAPS, 1612-1648)? If not or such data do not exist one cannot exclude a possibility that some of these CAIs (e.g., Type C-like) experienced melting in the chondrule-forming regions and therefore high Cr contents in CAI spinel do not reflect composition of their refractory precursors.
26. Line 261 It is difficult to infer formation age of CAIs from chemical compositions of CAI minerals

only. CAIs as a class of objects show large variations in volatility – from corundum-rich to forsterite-bearing.

Best wishes and Aloha,
Sasha Krot

Replies to the comments on the manuscript (NCOMMS-22-34336; “Chondrule-like objects and CAIs in asteroid Ryugu: earlier generations of chondrules”)

Note-1: The title of the paper is changed to “**Chondrule-like objects and CAIs in asteroid Ryugu: high temperature objects formed in the earliest stage of Solar System evolution**”, according to the comment 3-1.

Note-2: Modified phrases and sentences are highlighted with three different colors; yellow (Reviewer #1), blue (Reviewer #2), and green (Reviewer #3).

Note-3: Because of the limitation of the number of references, all the papers that the three reviewers introduced cannot be listed in the reference list.

Note-4: The followings are modifications based on the co-author’s comments (and my idea), but the conclusions of the present paper do not change.

4-1: One of the possible radial transport models mentioned in the final section, photophoresis (Moudens et al. 2011 A&A), is deleted. This model suggests that larger objects (~ 10 cm) are transported to farther locations rather than smaller objects (~ 10 μ m) when forming the Ryugu parent body (~ 2 Myr after CAI formation) and is not consistent with our observation, though none of the reviewers pointed out.

4-2: Figs. 3, 4, and 5 are modified so that our own data stand out; our data are changed from open symbols to filled symbols and literature data are changed from filled symbols to open symbols.

4-3: Two sentences in the Discussion section are modified to strengthen our arguments (L206-L208 and L337-L341), which are highlighted with gray.

Reviewer #1 (Remarks to the Author):

Comment 1-0: ...they are not really new as similar results were already published for both CI and Ryugu samples. I do not really understand why different working groups of the Hayabusa2 space mission try to publish similar data in different papers (this sounds like salami science to me and one single paper should be enough). In addition, I found that the interpretation is rather limited and not really put into perspective with the different (and some recent) models of CAI and chondrule formation.

Reply 1-0: To clarify of novelty of the present paper, blue sentence is added (L100-L101). Recent models and data are added in the Introduction and Discussion sections. See replies to the detailed comments.

Comment 1-1: Similar data have already been published for the CI chondrites Alais and Ivuna (Morin et al., 2022, GCA) and for Ryugu particles (Nakamura et al., 2022 Proc. Jpn. Acad., Ser.B 98). Although these papers are cited, the authors should report these data into the chemical and isotopic diagrams. In addition, the authors did not take into account the data reported for mechanically-extracted olivine grains from CI chondrites (Leshin et al., 1997, Piralla et al., 2020).

Reply 1-1: As the reviewer indicated, there are many oxygen isotope and elemental data of isolated olivine and pyroxene from the Ryugu samples and CI chondrites. Plotting all the oxygen isotope data makes Fig. 3 super-busy, and so oxygen isotope data only from the Ryugu samples are added to Fig. 3. According to this change, one sentence is added in the caption for Fig. 3 (L677-L679) and the relevant sentence is modified in the main text (L156-L157). The two papers (Leshin et al. 1997 GCA; Piralla et al. 2020 GCA) are mentioned in the main text (L85), and their data are plotted in Fig. 4 along with the Ryugu data (Liu et al., 2022 Nat. Astron.; Nakamura et al., 2022 Proc. Jpn. Acad., Ser.B 98). According to this change, the relevant sentence is modified in the caption for Fig. 4 (L687-L688) and one sentence is added in the main text (L194-L197).

Comment 1-2: In a general manner, the citations are very biased and mainly cite the work of the Wisconsin/American groups on chondrules, whereas other data exist, notably on the chemistry of relict olivines (see the work by the Nancy group). For instance, it has been shown that relict olivine grains in chondrules are Ca-Al-Ti depleted whereas host

olivines are not. This has been observed in both CC (Marrocchi et al., 2018, 2019, 2022), OC (Piralla et al., 2021) and R chondrites (Regnault et al., 2022). Why not comparing your data to this dataset for discussing your result?

Reply1-2: As for CaO-Cr₂O₃ plot in Fig. 4, type I chondrule-olivine data in type ≤ 3.0 chondrites reported in Marrocchi et al. (2018 EPSL, 2022 EPSL) and Piralla et al. (2021 GCA) are added to avoid the biased citations (L686). The olivine data from the CV chondrite Kaba (Marrocchi et al. 2019 GCA) have been excluded in the submitted version of the present paper, as the meteorite is petrologic type 3.1 (Bonal et al. 2006 GCA). The olivine data from R chondrites (Regnault et al. 2022 M&PS) are not used because the detailed petrologic types of the R chondrites are unknown.

The trend that relict olivine grains in chondrules are Ca-Al-Ti depleted whereas host olivine are not is mentioned in the main text (L180-L181).

Comment 1-3: Line 170. What do you mean by type < 3.0 chondrites? Types 2 and 1 that experienced aqueous alteration? If so, it does not sound like a good idea.

Reply 1-3: First, “type < 3.0 ” is not the precise phrase and changed to “type ≤ 3.0 ” so that type 3.0 (most primitive petrologic type) is included.

The relevant sentence is modified to explain why the olivine data is limited to type ≤ 3.0 chondrites (L182 – L185). Type ≤ 3.0 includes petrologic type 1 and 2, which experienced aqueous alteration. The Cr₂O₃ contents in olivine from aqueously-altered chondrites are as high as those from type 3.0 chondrites (Yamanobe et al. 2018 Polar Sci.; Schrader & Davidson 2017 GCA), which means aqueous alteration (~ 100 °C) does not affect Cr₂O₃ contents in olivine. But, this is not the case for type > 3.0 chondrites. The Cr₂O₃ contents in olivine decrease with increasing degree of petrologic type (Grossman & Brearley 2005 M&PS), which is due to Cr diffusion from olivine during thermal metamorphism.

Comment 1-4: It has been recently shown that small chondrules in CR chondrites are ¹⁶O-rich compared to larger chondrules (Marrocchi et al., 2022). In addition, a similar observation was reported long time ago by Bob Clayton for small chondrules in ordinary chondrites (Clayton et al., 1991). The authors should consider these studies for discussing their data. In addition, ¹⁶O-poor relict olivine grains compared to hosts have also been reported in CC chondrules (Marrocchi et al., 2018), thus confirming that both ¹⁶O-rich and ¹⁶O-poor isotopic reservoirs could have existed in the early solar system.

Reply 1-4: The hypothesis that smaller chondrules with relatively ¹⁶O-rich isotope ratios have escaped from incorporation into the ¹⁶O-poor CI-like dust (Marrocchi et al. 2022) is consistent with our interpretation. We added one sentence about their hypothesis (L236-L239).

The paper (Marrocchi et al. 2018 EPSL) is added as a reference to the sentence about existence of ¹⁶O-rich and -poor isotope reservoirs in the early stage of chondrule formation (L247).

Comment 1-5: Lines 211-216. This part shows a profound miss-conception of the SiO model proposed by Libourel, Dominik Hezel and the Nancy group. What appears unlikely is that SiO does not play a role in chondrule formation.

-First, we all agree that the Chaussidon et al. 2008 paper is wrong and there is no O-isotopic difference between host olivine and low-Ca pyroxene grains. It likely comes from a calibration problem of the matrix effect (no low-Ca pyroxene was available at this time in Nancy).

-The Marrocchi and Chaussidon 2015 paper is a model. As all models, it does not pretend to explain all the data. However, it shows that different dust/gas ratios can generate isotopic disequilibrium between olivine and low-Ca pyroxene (independently if it is observed or not in chondrules)

-Given that said, it does not exclude SiO as the ¹⁶O-poor endmember. Several papers recently reported complex textures in olivines of porphyritic chondrules, especially epitaxial growth of olivines in the outer part of chondrules (Libourel and Portail, 2018;

Libourel et al., 2022; Marrocchi et al., 2018, 2019; Jacquet et al., 2021, Piralla et al., 2021). These textures require the addition of Mg and SiO from the gas phase. If the authors think it can be achieved with water instead of Mg/SiO, they can propose an alternative.

-The Mg/SiO model does not predict that host olivine and low-Ca pyroxene grains to have different oxygen isotopic composition because both formed during gas-melt interactions. Water may have played a role in controlling the oxygen fugacity and part of the O-isotopic compositions but it appears extremely difficult to explain the textures without Mg/SiO (which also participate to the O isotopic budget).

-Claiming that Mg/SiO is unlikely is also difficult to reconcile with the presence of olivine and pyroxene (and even silica sometimes) in chondrules. How you will produce this mineralogical diversity without invoking SiO coming from the gas phase? If you only consider water, there is no reason you will crystalize olivine and pyroxene (and silica of course) in the same chondrule. In addition, low-Ca pyroxenes are currently observed in the outer part of chondrules, parallels to the surface. How can explain that without an increase of the silicon activity in chondrules?

-Another isotopic constraint that supports Mg/SiO is the large Si isotopic compositions observed in type I porphyritic chondrules, which is impossible to explain without SiO. Hence, there are a lot of evidence of the role played by Mg/SiO during chondrule formation. Trying to dismiss this model with only one weak argument is not serious considering the large dataset available in literature (which you did not take into account).

Reply 1-5: First of all, the discussion whether the ^{16}O -poor counterpart is chondrule precursor dust or SiO gas is not the main theme of the present paper, and our own data in the present study are not able to give an answer to this issue. So, based on the previous oxygen isotope studies summarized in Tenner et al. (2018), we suggested that the ^{16}O -poor counterpart is chondrule precursor dust but not SiO gas. Now the reviewer indicated the oxygen isotope fractionation between olivine and pyroxene in chondrules, which is the argument for the SiO model, is an analytical artifact. So, we delete the counterargument against the SiO model based on the oxygen isotope ratios in chondrules.

Another counterargument against the SiO model is described in Chaumard et al. (2021 GCA). According to this paper, we discuss whether the ^{16}O -poor counterpart is chondrule precursor dust or SiO gas and suggest SiO gas is less likely (L239-L243).

The reviewer seems misunderstanding. In the present paper, we do not suggest H_2O as the ^{16}O -poor counterpart but suggest chondrule precursor dust as the ^{16}O -poor counterpart. The ^{16}O -poor H_2O serves as an oxidant for forming ^{16}O -poor type II chondrules from relatively ^{16}O -rich type I chondrule-like precursors (e.g., Tenner et al., 2018). It would be possible to form pyroxene via mixing between chondrule precursor dust with CI-like compositions and olivine-rich chondrule-like objects. Marrocchi et al. (2022 EPSL) suggested that ^{16}O -poor chondrule precursor dust was added to form ^{16}O -poor chondrules from relatively ^{16}O -rich chondrules. This means chondrule precursor dust can be the ^{16}O -poor counterpart.

Comment 1-6: Lines 284-286. There is a current debate today if chondrules accreted where they formed or not (Williams et al., 2020 vs. Schneider et al. 2020 for instance). If your data (now) support no transport, it would be great to explain this debate and the different model that have been proposed. This can be tested by plotting the $\Delta^{17}\text{O}$ versus $\Delta^{18}\text{O}$ ($\delta^{18}\text{O}-\delta^{17}\text{O}$). It has been recently shown that there is a difference between chondrules formed in the inner part and the outer part of the disk using this diagram (Piralla et al., 2021).

Reply 1-6: The relevant sentence is modified to include the recent papers discussing migration of chondrules (L317-L318).

Comment 1-7: The discussion on why CAIs are observed in CI/Ryugu samples does not take into account the recent models where CAIs could have been formed further away from the Sun than previously thought, during the disk viscous spreading.

For instance, the dissipation of turbulence could sustain CAI-forming temperatures at distances of several tenths of an AU, especially at early times during disk build-up where the massive disk could retain heat more efficiently (Hueso and Guillot, 2005; Dullemond et al., 2006; Yang and Ciesla, 2012; Pignatale et al., 2018). In addition, the formation of CAIs/AOAs at larger heliocentric distances is now supported by recent geochemical measurements (V isotopes) suggesting that CAIs experienced low levels of irradiation that are difficult to reconcile with their formation at the disk inner edge (Bekaert et al., 2021). In addition, carbonaceous chondrites, including CIs, contain the most refractory inclusions of any chondrite class, even though they are thought to have accreted in the outer solar system at <200 K (Warren, 2011). This paradox can be mitigated if (i) refractory inclusions were entrained by the initial outward expansion of the disk (e.g., Pignatale et al. 2018)

and/or (ii) preferentially trapped at the outer edge of the gap carved by growing Jupiter (e.g., Desch et al., 2018).

Reply 1-7: The papers that the reviewer introduced suggest that CAIs have formed at greater heliocentric distance (e.g., ~ 1 au; Baekart et al., 2021 *Sci. Adv.*) than previously thought (~ 0.1 au), but not as far as formation locations of the Ryugu original parent body and comet Wild 2. The introduced papers suggested radial transport of CAIs after formation. So, the relevant sentences are modified to include the hypothesis of the CAI formation at further locations from the Sun than previously considered (**L57** and **L303**). Due to the limitation of number of citations (up to 70 in the main text, probably), all the papers that the reviewer introduced cannot be referred.

Comment 1-8: A CAI has also been reported in the CI Ivuna and should be mentioned (Frank et al., 2011, 2017).

Reply 1-8: A sentence about discovery of a CAI in the Ivuna CI chondrite is added (**L85-L86**).

Reviewer #2 (Remarks to the Author):

Comment 2-1: The manuscript needs to include and would benefit from an acknowledgement and comparison to agglomeratic olivine (AO) chondrules that we found in the Renazzo-like carbonaceous (CR) chondrites. We demonstrated that these AO chondrules were indistinguishable (size, texture, chemical, and O-isotopic composition) from the chondrule-like objects found in Comet Wild 2 (Schrader et al., 2018-GCA; <https://doi.org/10.1016/j.gca.2017.12.014>). The objects we found in CR chondrites have the same size and O-isotopic composition as those objects seen by these authors in Ryugu, and we argued for a similar origin from amoeboid olivine aggregates very early in the Solar system, and outward migration of these objects well.

Schrader D. L., Nagashima K., Waitukaitis S. R., Davidson J., McCoy T. J., Connolly Jr. H. C., and Lauretta D. S. (2018) The retention of dust in protoplanetary disks: Evidence from agglomeratic olivine chondrules from the outer Solar System. *Geochim. Cosmochim. Acta* 223, 405–421.

Relevant citation locations/areas to address the results of Schrader et al. (2018): Lines 74; 117/118+; 166/167; 183 – 190; ‘Earlier Generation of Chondrules’ Section starting at Line 192; 221; 280 – 283. Could also add EPMA data from Schrader et al. (2018) on AO chondrules to Fig. 4 (CaO vs. Cr₂O₃ wt.%).

Reply 2-1: We added sentences that compare the chondrule-like objects and the AO chondrules (**L229-L232**). Fig. 4 is used for classification of the chondrule-like objects (AOA-like or typical chondrule-like). The AO chondrules are not typical chondrules and certainly contain relict AOA-olivine, so that olivine data of the AO chondrules are not suitable to be plotted in Fig. 4.

Comment 2-2: “...resembling what proposed as earlier generations of chondrules”
-Change to: resembling what has been proposed as earlier generations of chondrules?

Reply 2-2: Modified (**L38** and **L221**).

Comment 2-3: I think Comet Wild 2 should have a space between Wild and 2. Wild2 is used throughout the current draft. See Zolensky et al. (2006-Science; doi: 10.1126/science.1135842).

Reply 2-3: A space between Wild and 2 is added (L63 and other places).

Comment 2-4: Lines 88-89. "...which is as early as formation of major types of carbonaceous chondrite chondrules at 2.2 – 2.7 Myr after CAI formation..."

-Since this isn't the age range of chondrule formation for all types of carbonaceous chondrites (i.e., CR chondrules formed later), I recommend being specific about which meteorite types are meant here.

Reply 2-4: The phrase is modified as "... which is as early as formation of chondrules from major types of carbonaceous chondrites such as CM, CO, and CV at 2.2 – 2.7 Myr after CAI formation" (L96-L98 and L323-L325).

Comment 2-5: Update reference #54 from the LPSC abstract to the very recently published paper.

Liu et al. (2022) Incorporation of ¹⁶O-rich anhydrous silicates in the protolith of highly hydrated asteroid Ryugu, Nature Astronomy. <https://www.nature.com/articles/s41550-022-01762-4>

Reply 2-5: Modified (reference #39).

Reviewer #3 (Remarks to the Author):

Comment 3-0: ... but not without caveat: chronology of objects cannot be uniquely inferred from the data obtained. It is a pity that not attempts were made for measuring Al-Mg systematics in hibonite of one of the CAIs described. It may be too difficult considering small grain sizes or it may be a goal of future studies. These are still relatively minor issues and ...

Reply 3-0: We wanted to measure Al-Mg isotope systematic of hibonite and spinel in the Ryugu CAIs, for which technical developments that enable Al-Mg isotope analysis with 1 μm spots are required. But the Ryugu samples had to be sent back to ISAS/JAXA in this past June. Because of this time limitation, we postponed the Al-Mg isotope analysis. Instead of the Al-Mg chronometry, we used chemical compositions of the Ryugu CAIs for rough estimation of the formation ages.

Detailed comments:

Comment 3-1: The title does not exactly reflect the content of the paper. Although the CAIs in the Ryugu samples are possibly among the oldest refractory inclusions (the expectation is that these CAIs are ²⁶Al-poor), there is no chronological information in the paper. In addition, igneous CAIs cannot be considered to be earlier generations of chondrules: chondrules and igneous CAIs were melted in different disk regions characterized by different O-isotope compositions and ambient temperatures.

Reply 3-1: According to this comment. We changed the title of the present paper. See Note-1.

Comment 3-2: Line 56 – Please add U-corrected Pb-Pb ... and add a reference to Connelly et al. (2012) Science

Reply 3-2: Modified (L58) and added the reference (#9).

Comment 3-3: AOAs are aggregates of solar nebular condensates. Since forsterite has lower condensation temperatures than most minerals in CAIs, AOAs represent lower temperature condensates than CAIs.

Reply 3-3: According to this comment, the relevant sentence is modified (L59-L60).

Comment 3-4: Oxygen-isotope (please use hyphen) & oxygen isotopic (no hyphen)

Reply 3-4: Modified (L67 and other places).

Comment 3-5: Line 65 – Chondrules are isotopically uniform only in unmetamorphosed (petrologic types < 3.0). It may be better to define $\delta^{17}\text{O}$ and $\delta^{18}\text{O}$ and use them in the text instead of “oxygen-isotope ratios” which are too often used in lines 65-71.

Reply 3-5: As for the first sentence in this comment, the relevant sentence is modified (L70-L72). The definition of $\delta^{17}\text{O}$ and $\delta^{18}\text{O}$ values is added, and the relevant sentences are modified (L69-L70). The phrase “oxygen isotope” is changed to “oxygen-isotope” or “ $\delta^{18}\text{O}$ and $\delta^{17}\text{O}$ values”.

Comment 3-6: Line 68 – It may be better to use PCM line instead of “slope-1 line”. Note that PCM is used in line 148.

Reply 3-6: Modified (L68-L69).

Comment 3-7: Line 75 - ... phyllosilicates

Reply 3-7: Modified (L82 and other places).

Comment 3-8: Line 76 - ... carbonates

Reply 3-8: Modified (L83)

Comment 3-9: Line 78 Frank et al. described polymineralic CAI in CI chondrite. It is uniformly ^{16}O -rich and quite refractory.

Reply 3-9: See Reply 1-8.

Comment 3-10: Line 90-91 – Are the Ryugu samples or CAIs and chondrules smaller than 30 μm ?

Reply 3-10: To avoid the confusion, the sentence is modified (CAIs and chondrule-like objects are small) (L98-L100).

Comment 3-11: Line 92 - Not all elements analyzed are major.

Reply 3-11: Deleted “major” (L103 and L365-L366).

Comment 3-12: Line 96 – Delete “As a result”. You may say “Our studies indicate that ...”

Reply 3-12: Modified (L106-L107).

Comment 3-13: Line 110 – You may need to mention that chondrule-like objects and CAIs were discovered in C samples which are less altered than A samples.

Reply 3-13: As the reviewer pointed out, the chondrule-like objects and CAIs were discovered in C0002, C0040, and C0076 but not from A samples. But it is difficult to say that C samples are less altered than A samples from the limited number of the discoveries. The difference in degree of aqueous alteration between A and C samples will be discussed in a separated paper.

Comment 3-14: Line 117 - I am not sure if it is allowed to refer to a submitted paper.

Reply 3-14: The paper “Nakamura T. et al. 2022 Science” (#28) has been accepted in this past September. If a paper under review cannot be referred in the Nature Communications paper, Kawasaki et al. (2022) (#50) has to be deleted, because the paper is under review as far as I know.

Comment 3-15: Line 134 – it may be better to use “inclusions” instead of “particles”.

Reply 3-15: Modified (L143).

Comment 3-16: Please use “phyllosilicates” in the text.

Reply 3-16: Modified (L82 and other places).

Comment 3-17: Line 183 – Some AOAs are compact coarse-grained objects containing subhedral-to-euhedral diopside grains; they appear to have experienced melting.

Reply 3-17: According to this comment, the relevant sentence is modified (L201-L202).

Comment 3-18: Line 193 – I don't know chondrules having ^{16}O -rich isotopic compositions except a single chondrule described by Kabayashi et al.

Reply 3-18: There has been no chondrule as ^{16}O -rich as (or more ^{16}O -rich than) CAIs and AOAs, except for the chondrule reported in Kobayashi et al. (2003 *Geochem. J.*).

Comment 3-19: Line 199 - ... chondrule-like ...

Reply 3-19: Modified (L219).

Comment 3-20: Note that igneous CAIs and AOAs (some AOAs have been melted) experienced annealing and contain no glass, but we do not call them early generations of chondrules. Same comment for line 211.

Reply 3-20: We agree to this comment and changed the title of the present paper to avoid misleading that both the chondrule-like objects and CAIs are earlier generations of chondrules. See Note-1.

Comment 3-21: Subsequent TEM studies showed that objects described by Libourel and Krot as “fragments of differentiated planetesimals” in fact contain glass along “dry contacts”, i.e. they were incorrectly interpreted. It may be unnecessary to discuss in detail their paper.

Reply 3-21: The TEM studies cannot be found. But our argument is not changed with/without the TEM studies, so we keep the relevant sentences as they are.

Comment 3-22: The ^{16}O -poor counterpart is dust dominated the chondrule-forming regions characterized by high dust/gas ratios. The SiO gas results from evaporation of such disk regions.

Reply 3-22: We agree to this comment. See reply 1-5.

Comment 3-23: Line 217-219. The co-existence of ^{16}O -rich and ^{16}O -depleted gaseous reservoirs during melting of refractory dust has been inferred in several papers (e.g., Kawasaki et al., 2018 and references therein).

Reply 3-23: The paper by Kawasaki et al. (2018) is added as a reference in the sentence (L247).

Comment 3-24 (Line 244): It is not clear why cometary CAIs are younger than CM CAIs. Is it based on Al-Mg isotope systematics? Note that CM PLACs have no radiogenic ^{26}Mg excess whereas SHIBs have ^{26}Mg excess corresponding to the canonical $^{26}\text{Al}/^{27}\text{Al}$ ratio.

Reply 3-24: As the reviewer pointed out, PLACs (or hibonite-rich CAIs) are products before the ^{26}Al injection into the solar nebula, so that they show no ^{26}Mg excess and are apparently young. But, the cometary CAIs are mineralogically different from the hibonite-rich CAIs. Therefore, the cometary CAIs are most likely young. Sentences about this argument are added (L286-L290).

Comment 3-25: Line 245-246. Do all cometary CAIs measured for O-isotope composition have uniform ^{16}O -rich compositions (it seems like this is not the case – see Fig. 13 in Joswiak et al. 2017, *MAPS*, 1612-1648)? If not or such data do not exist one cannot exclude a possibility that some of these CAIs (e.g., Type C-like) experienced melting in the chondrule-forming regions and therefore high Cr contents in CAI spinel do not reflect composition of their refractory precursors.

Reply 3-25: The reviewer means that acquisition of Cr by the remelted CAIs occurred in the chondrule-forming regions. The phrase “in the chondrule-forming regions” is added to the relevant sentence (L284).

Comment 3-26: Line 261 It is difficult to infer formation age of CAIs from chemical compositions of CAI minerals only. CAIs as a class of objects show large variations in volatility – from corundum-rich to forsterite-bearing.

Reply 3-26: I agree with the reviewer's comment. So, the relevant sentence is modified

to be more softened (**L292**).

REVIEWER COMMENTS

Reviewer #1 (Remarks to the Author):

This is my second review of this article. I still do not really understand why similar results from the Ryugy samples should be published in 3 different papers (Liu et al., 2022, Nakamura et al., 2022). This is clearly oversold regarding the importance of the discovery. Given that said, the authors took into account the reviewers' comment and made the job for modifying the paper (except for SiO again...). Here some comments and suggestions.

1- I do not really understand what the authors proposed for forming chondrules from earlier generation of chondrules. Do they propose interactions with water? If so, they should detail how they can explain the peculiar textures of porphyritic type I chondrules, characterized by olivine showing variable Ca-Al-Ti contents and epitaxial growth. A comprehensive model would be welcome.

2- Page 5, line 100. This is not the first report (see Liu et al., 2022, Nakamura et al., 2022).

3- Page 6, line 134. 120° triple junctions are not only produced by annealing but can also correspond to epitaxial growth during gas-melt interactions (see the textures reported by Libourel and Portail 2018, Science Advances, and Marrocchi et al. 2018, EPSL).

4- Page 9, line 218. Indeed, the fact that chondrule-like objects showing triple junction cannot be products from differentiated planetesimals is supported by oxygen isotopes. This was shown in Marrocchi et al. 2018 (EPSL), which should be cited.

5- Page 10, lines 236-243. Again, there is a fundamental miss-understanding of the SiO model. Trying to discredit this model would require more arguments than the really weak (and wrong) argument proposed in this paper.

First, there is no reference to SiO in the paper by Marrocchi et al., 2022. The idea of the model is that the larger CR chondrules derived from small ones by the addition of CI-like matrix. I would be pleased to know how the authors would explain the peculiar ^{54}Cr isotopic compositions of CR chondrules by only considering water during chondrule formation? In addition, mass balance calculations regarding Cr concentrations of CR chondrules suggest that CI-like dust should be water-free. If the authors want to discard this model, they should quantitatively explain why and proposed an alternative explaining the CR chondrule textures as well as chemical and isotopic characteristics. I will be happy to read it...

Second, the authors proposed that the partial pressure of SiO is low due to the low temperature of the disk when chondrules formed. Where comes from the idea that SiO should come from the evaporation of chondrule melt? Please give references for that but I doubt you will find some.

Third, the authors proposed that the partial pressure of SiO is low because the temperature of the disk was low during chondrule formation. Chondrules are magmatic objects. They formed at temperature higher than 1400°C (see Schnuriger et al., 2022, MAPS, for an estimate of the temperature at which olivine formed in chondrules). At such temperature, the gas chondrules interacted with is thus composed of different species, for instance: water (I have no problem with water being present) and SiO (the more stable carrier of Si in the disk).

I strongly suggest to remove the SiO discussion as this clearly becomes ridiculous to attack this model by erroneously citing the articles and without providing a quantitative and comprehensive alternative. Water was present during chondrule formation and likely controlled the redox state of these objects (no doubt about that). However, the peculiar textural characteristics of chondrules cannot be only explained by water (strangely, the authors never refer to these textures). Same goes for oxygen and ^{54}Cr isotopes.

6- Page 13, lines 315-317. Cite also Schneider et al. 2020 (EPSL) as they proposed that chondrules

agglomerated where they formed (based on ^{54}Cr and ^{50}Ti).

7- Page 13, line 331. Morin et al 2022 GCA proposed that CI olivine grains could derive chondrules formed in the outer part of the disk due to their similarity with CR chondrules (so specify this). In addition, larger olivine grains are also present in CI and only limited to olivines smaller than 30 microns (see Leshin et al., 1997, Piralla et al., 2020).

Reviewer #3 (Remarks to the Author):

The paper provides a comprehensive study of the unique high-temperature primary objects, CAIs and chondrules, in the extremely precious samples returned by JAXA Hayabusa mission from the Ryugu asteroid. Several teams have been working on small particles from 5.4 gram samples available. The distribution of high temperature primary solids in these particles is very heterogeneous; most particles consist entirely of low temperature secondary minerals. The authors reported ^{16}O -rich chondrule-like objects not previously described in the literature. The CAIs show clear evidence for low temperature aqueous alteration indicative of their accretion to the Ryugu parent asteroid prior to its aqueous alteration. This very important signature has not been described either. The data reported in the paper are of high quality and great interest to the cosmochemical community. Therefore I highly recommend publication of this manuscript in Nature Communication after minor revision (see my detailed comments below).

Detailed comments:

When one uses “earlier generations of chondrules” a comparison is required, e.g., “than typical chondrules in carbonaceous chondrites”. If there is no such comparison, one should use “early or earliest generations of chondrules”. Please correct throughout the text.

Line 41. ... as old as the oldest CAIs – Strictly speaking, there are no strong arguments supporting this statement.

Line 42 ... from comets – Do you mean Wild 2 comet?

Line 45 ... alteration on the Ryugu parent asteroid.

Line 51. Delete “,”

Line 57. ... in a gas of approximately solar ...

Line 61. ... anorthite; they are as old as ...

Line 64. This statement is correct for CAIs but not for chondrules. Chondrule formation occurred throughout the protoplanetary disk, inside and outside Jupiter.

Line 71. ... in individual chondrules ...

Line 75-77. Chondrules in CCs plot below the TF line along the PCM line. Chondrules in OCs plot above the TF line, but not along the PCM line.

Line 80. From this sentence it is unclear whether genetic link is suggested for relict grains or coexisting mineral phases (why not to call them phenocrysts).

Line 84. You may delete “in the CI chondrites”.

Line 85 to avoid second “observed”, please use “Frank et al. described ...”

Line 85-88. It may be better to move the sentence Frank et al... to the end of this paragraph.

Line 113-116. Move first sentence after “... in total).”

Line 129. Fe-Ni metal and sulfide are present in two of them.

Line 130. ... contains Al- and Ti-free diopside

Line 141. Delete “in the present study”.

Line 144-145. Phyllosilicates ... are ... Please check the rest of the text. This error appears in many places.

Line 147 Phyllosilicates ... have ...

Line 154 ... more complete information ...

Line 158. The individual objects are isotopically uniform with the uncertainty of our measurements.

Line 161. ... spot); the latter contains ...

Line 161. The third object

Line 184 .. are preserved undisturbed only in ...

Line 187. Note that Cr is rather common in Fe,Ni-metal in type I chondrules. In addition, Cr content is rather high in type II chondrules (they often contain chromite).

Line 197. Reference is required.

Line 198. ... in the anomalously ¹⁶O-rich chondrule ...

Line 214. ... are observed as relict grains in chondrules ...

Line 240-243. SiO gas in the chondrule-forming regions resulted from evaporation of ¹⁶O-depleted dust and water ice. There is no contradiction between (1) incorporation of ¹⁶O-rich chondrules into ¹⁶O-depleted dust followed by subsequent melting and (2) melting of ¹⁶O-rich chondrules in an ¹⁶O-depleted gas produced by evaporation of ¹⁶O-depleted dust.

Line 259-260. I thought that primary minerals in CAIs are embedded in phyllosilicates which probably replaced melilite. Here you wrote that CAIs are surrounded by phyllosilicates, which must be incorrect (melilite cannot be outside CAIs).

Line 268. ... by alteration phases, like ..., and spinel ... phases, like ..., are observed ...

Line 271. Could you please explain why cometary CAIs are younger than CM CAIs.

Line 274. If CAIs were remelted during chondrule formation (rather common situation for Type C CAIs), they must be considered as chondrules formed by melting of CAI-like precursors.

Line 287. ... apparently ... Do you mean "appear to be"?

Line 298. .. smallness ... - small sizes ...

Replies to the comments on the manuscript (NCOMMS-22-34336A; “Chondrule-like objects and CAIs in asteroid Ryugu: high temperature objects formed in the earliest stage of Solar System evolution”)

Note: Modified phrases and sentences are highlighted with two different colors; yellow (Reviewer #1) and green (Reviewer #3).

Reviewer #1 (Remarks to the Author):

Comment 1-1: I do not really understand what the authors proposed for forming chondrules from earlier generation of chondrules. Do they propose interactions with water? If so, they should detail how they can explain the peculiar textures of porphyritic type I chondrules, characterized by olivine showing variable Ca-Al-Ti contents and epitaxial growth. A comprehensive model would be welcome.

Reply 1-1: As we answered in the previous replies, we do not suggest H₂O as the ¹⁶O-poor counterpart but suggest chondrule precursor dust as the ¹⁶O-poor counterpart (actually the word “water” does not appear in the main text, figures, and table). The ¹⁶O-poor H₂O serves as an oxidant for forming ¹⁶O-poor type II chondrules from relatively ¹⁶O-rich type I chondrule-like precursors (e.g., Tenner et al., 2018).

The chondrule-like objects in the Ryugu samples resemble olivine-metal-aggregates with triple junctions that have been proposed as earlier generations of chondrules (Libourel and Krot, 2007), and the chondrule-like objects probably escaped from incorporation into the ¹⁶O-poor chondrule precursor dust followed by reheating events that results in forming chondrules with/without ¹⁶O-rich relict olivine grains. That is what we argue in the discussion.

Comment 1-2: Page 5, line 100. This is not the first report (see Liu et al., 2022, Nakamura et al., 2022).

Reply 1-2: The relevant sentence is modified to clarify what is new in the present study (L100-101). The two papers that the reviewer mentioned reported AOA-like objects and isolated anhydrous grains in the Ryugu samples.

Comment 1-3: Page 6, line 134. 120° triple junctions are not only produced by annealing but can also correspond to epitaxial growth during gas-melt interactions (see the textures reported by Libourel and Portail 2018, Science Advances, and Marrocchi et al. 2018, EPSL).

Reply 1-3: One sentence about epitaxial growth of olivine in chondrules is added (L138-139). Even with this modification, our argument does not change. The ¹⁶O-rich chondrule-like object with 120° triple junctions share characteristics with AOAs.

Comment 1-4: Page 9, line 218. Indeed, the fact that chondrule-like objects showing triple junction cannot be products from differentiated planetesimals is supported by oxygen isotopes. This was shown in Marrocchi et al. 2018 (EPSL), which should be cited.

Reply 1-4: Marrocchi et al. 2018 (EPSL) is added to the relevant sentence as a reference (L228-229).

Comment 1-5: Page 10, lines 236-243. Again, there is a fundamental miss-understanding of the SiO model. Trying to discredit this model would require more arguments than the really weak (and wrong) argument proposed in this paper.

First, there is no reference to SiO in the paper by Marrocchi et al., 2022. The idea of the model is that the larger CR chondrules derived from small ones by the addition of CI-like matrix. I would be pleased to know how the authors would explain the peculiar ⁵⁴Cr isotopic compositions of CR chondrules by only considering water during chondrule formation? In addition, mass balance calculations regarding Cr concentrations of CR chondrules suggest that CI-like dust should be water-free. If the authors want to discard this model, they should quantitatively explain why and proposed an alternative explaining

the CR chondrule textures as well as chemical and isotopic characteristics. I will be happy to read it...

Second, the authors proposed that the partial pressure of SiO is low due to the low temperature of the disk when chondrules formed. Where comes from the idea that SiO should come from the evaporation of chondrule melt? Please give references for that but I doubt you will find some.

Third, the authors proposed that the partial pressure of SiO is low because the temperature of the disk was low during chondrule formation. Chondrules are magmatic objects. They formed at temperature higher than 1400°C (see Schnuriger et al., 2022, MAPS, for an estimate of the temperature at which olivine formed in chondrules). At such temperature, the gas chondrules interacted with is thus composed of different species, for instance: water (I have no problem with water being present) and SiO (the more stable carrier of Si in the disk).

I strongly suggest to remove the SiO discussion as this clearly becomes ridiculous to attack this model by erroneously citing the articles and without providing a quantitative and comprehensive alternative. Water was present during chondrule formation and likely controlled the redox state of these objects (no doubt about that). However, the peculiar textural characteristics of chondrules cannot be only explained by water (strangely, the authors never refer to these textures). Same goes for oxygen and ^{54}Cr isotopes.

Reply 1-5: First of all, the SiO discussion is removed. The conclusions are not changed even with this modification. The model in Marrocchi et al. (2022 EPSL) that ^{16}O -rich objects (or chondrules) were incorporated into ^{16}O -poor dust and reheated is consistent with the idea in the present paper, so that the relevant sentence is kept as it is (L238-L241).

Regarding the comment about Cr isotopes in chondrules, we do not consider H_2O as the ^{16}O -poor counterpart to the ^{16}O -rich chondrule-like objects, as mentioned in Reply 1-1.

There are papers discussing SiO evaporation from chondrule melt based on experiments (e.g., Hashimoto, 1983 *Geochem. J.*; Imae & Isobe, 2017 EPSL) and theoretical calculation (Fedkin et al., 2006 GCA) See also the review by Davis et al. (2005 *In Chondrites and Protoplanetary Disk*). So, SiO evaporation from chondrule melt is not a new idea. The papers mentioned here will not be cited in the present paper, as the SiO discussion is removed.

Comment 1-6: Page 13, lines 315-317. Cite also Schneider et al. 2020 (EPSL) as they proposed that chondrules agglomerated where they formed (based on ^{54}Cr and ^{50}Ti).

Reply 1-6: Schneider et al. 2020 (EPSL) is added to the relevant sentence as a reference (L318).

Comment 1-7: Page 13, line 331. Morin et al 2022 GCA proposed that CI olivine grains could derive chondrules formed in the outer part of the disk due to their similarity with CR chondrules (so specify this). In addition, larger olivine grains are also present in CI and only limited to olivines smaller than 30 microns (see Leshin et al., 1997, Piralla et al., 2020).

Reply 1-7: The relevant sentence is modified (... the limited size range of the “most” isolated grains...) (L333). If comparing the size of isolated olivine grains between CI chondrites and other carbonaceous chondrites on the same condition of observation on polished sections, isolated olivine grains in CI chondrites are smaller than those in other carbonaceous chondrites (< 25 μm in Alfing et al., 2019 *Geochemistry* and < 30 μm in Morin et al., 2022 GCA vs. up to 200 μm in Jacquet et al., 2021 M&PS). The two papers that the reviewer mentioned (Leshin et al., 1997; Piralla et al., 2020) reported olivine grains extracted by mechanical separation, which are not suitable for comparison. If the isolated olivine grains formed in the CI chondrite formation regions as suggested in Morin et al. (2022), explanation of the difference in size of isolated olivine grains between CI chondrites and other carbonaceous chondrites is needed. Rather than in-situ formation of isolated olivine grains, we argue selective transport favoring smaller olivine grains to the

formation location of the Ryugu parent body which is farther from the Sun than any other chondrite parent bodies.

Reviewer #3 (Remarks to the Author):

Comment 3-0: When one uses “earlier generations of chondrules” a comparison is required, e.g., “than typical chondrules in carbonaceous chondrites”. If there is no such comparison, one should use “early or earliest generations of chondrules”. Please correct throughout the text.

Reply 3-0: We changed the wording from “earlier generations of chondrules” to “early generations of chondrules” (highlighted with green). But the wordings in **L231** (AO chondrules as earlier generations of chondrules) and **L235** (¹⁶O-rich relict olivine as earlier generations of chondrules) are kept as they are with respecting the views in Schrader et al. (2018 GCA) and Marrocchi et al. (2019 EPSL).

Comment 3-1: Line 41. ... as old as the oldest CAIs – Strictly speaking, there are no strong arguments supporting this statement.

Reply 3-1: If we clearly say the two Ryugu CAIs are as old as the oldest CAIs, Al-Mg isotope chronometry is needed. But there is the observation that spinel in older CAIs contains lower Cr₂O₃ concentrations, and spinel in the two Ryugu CAIs contains Cr₂O₃ as low as those in spinel in oldest CAIs, though this is not strong argument. That is why we put “possibly” before “as old as the oldest CAIs”.

Comment 3-2: Line 42 ... from comets – Do you mean Wild 2 comet?

Reply 3-2: Not only Wild 2 but also other comets. Chondrule-like fragments in cometary IDPs are as small as chondrule-like Wild 2 particles. One sentence about this observation is added in **L307-309**, not in abstract.

Comment 3-3: Line 45 ... alteration on the Ryugu parent asteroid.

Reply 3-3: Modified (**L45**). The abstract exceeded 150 words by this modification (155 words). The guideline of Nature Communications says “an abstract of approximately 150 words”, and so I hope 155 words are ok.

Comment 3-4: Line 51. Delete “,”

Reply 3-4: Deleted (**L51**).

Comment 3-5: Line 57. ... in a gas of approximately solar ...

Reply 3-5: Modified (**L57**).

Comment 3-6: Line 61. ... anorthite; they are as old as ...

Reply 3-6: Modified (**L61-62**).

Comment 3-7: Line 64. This statement is correct for CAIs but not for chondrules. Chondrule formation occurred throughout the protoplanetary disk, inside and outside Jupiter.

Reply 3-7: I agree with the comment. But the important point here is CAIs and chondrules are existed throughout the protoplanetary disk, regardless of the original formation locations.

Comment 3-8: Line 71. ... in individual chondrules ...

Reply 3-8: Modified (**L71**).

Comment 3-9: Line 75-77. Chondrules in CCs plot below the TF line along the PCM line. Chondrules in OCs plot above the TF line, but not along the PCM line.

Reply 3-9: No, there are chondrules in CCs of which oxygen isotope ratios plot above the TF line along the PCM line, e.g., FeO-rich type II chondrules in CR chondrites and chondrules in CH and CB chondrites (see Tenner et al., 2018). So, we do not change the

relevant sentence.

Comment 3-10: Line 80. From this sentence it is unclear whether genetic link is suggested for relict grains or coexisting mineral phases (why not to call them phenocrysts).

Reply 3-10: The relevant phrase is modified to clarify the genetic link of relict grains to CAIs and AOAs (L80-81).

Comment 3-11: Line 84. You may delete “in the CI chondrites”.

Reply 3-11: Deleted (L83-84).

Comment 3-12: Line 85 to avoid second “observed”, please use “Frank et al. described ...”

Reply 3-12: Modified (L88).

Comment 3-13: Line 85-88. It may be better to move the sentence Frank et al... to the end of this paragraph.

Reply 3-13: Moved and the sentence is slightly modified according to the move (L87-88).

Comment 3-14: Line 113-116. Move first sentence after “... in total).”

Reply 3-14: Modified (L115-116).

Comment 3-15: Line 129. Fe-Ni metal and sulfide are present in two of them.

Reply 3-15: Modified (L129).

Comment 3-16: Line 130. ... contains Al- and Ti-free diopside

Reply 3-16: Modified (L130).

Comment 3-17: Line 141. Delete “in the present study”.

Reply 3-17: Deleted (L142).

Comment 3-18: Line 144-145. Phyllosilicates ... are ... Please check the rest of the text. This error appears in many places.

Reply 3-18: Modified (L144-145). The errors in other places are also modified (highlighted with green).

Comment 3-19: Line 147 Phyllosilicates ... have ...

Reply 3-19: Modified (L147-148).

Comment 3-20: Line 154 ... more complete information ...

Reply 3-20: Modified (L155).

Comment 3-21: Line 158. The individual objects are isotopically uniform with the uncertainty of our measurements.

Reply 3-21: Modified (L158-159).

Comment 3-22: Line 161. ... spot); the latter contains ...

Reply 3-22: Modified (L162).

Comment 3-23: Line 161. The third object

Reply 3-23: Modified (L162).

Comment 3-24: Line 184 .. are preserved undisturbed only in ...

Reply 3-24: Modified (L185-186).

Comment 3-25: Line 187. Note that Cr is rather common in Fe,Ni-metal in type I chondrules. In addition, Cr content is rather high in type II chondrules (they often contain chromite).

Reply 3-25: As the reviewer pointed out, type II chondrules contain higher Cr than type I chondrules (e.g., Kita et al., 2010 GCA). We compare CaO and Cr₂O₃ concentrations in olivine between the chondrule-like objects and type-I chondrules (and AOAAs) (Fig. 4), but not type-II chondrules. As the reviewer mentioned, Fe-Ni metal in type-I chondrules contain certain amount of Cr. But what we discuss is Cr contents in olivine. Olivine/melt partition coefficient of Cr is about 1 (Kennedy et al., 1993 EPSL), but Cr is originally minor in chondrule precursor dust (Tenner et al., 2018 In Chondrules). Therefore, Cr₂O₃ concentrations in type-I chondrule olivine are low.

Comment 3-26: Line 197. Reference is required.

Reply 3-26: References are added (L198).

Comment 3-27: Line 198. ... in the anomalously ¹⁶O-rich chondrule ...

Reply 3-27: Modified (L199).

Comment 3-28: Line 214. ... are observed as relict grains in chondrules ...

Reply 3-28: Modified (L215).

Comment 3-29: Line 240-243. SiO gas in the chondrule-forming regions resulted from evaporation of ¹⁶O-depleted dust and water ice. There is no contradiction between (1) incorporation of ¹⁶O-rich chondrules into ¹⁶O-depleted dust followed by subsequent melting and (2) melting of ¹⁶O-rich chondrules in an ¹⁶O-depleted gas produced by evaporation of ¹⁶O-depleted dust.

Reply 3-29: The discussion about SiO gas as the ¹⁶O-poor counterpart is now deleted. See Reply 1-5.

Comment 3-30: Line 259-260. I thought that primary minerals in CAIs are embedded in phyllosilicates which probably replaced melilite. Here you wrote that CAIs are surrounded by phyllosilicates, which must be incorrect (melilite cannot be outside CAIs).

Reply 3-30: The incorrect description is modified (L257-258).

Comment 3-31: Line 268. ... by alteration phases, like ..., and spinel ... phases, like ..., are observed ...

Reply 3-31: Modified (L267-268).

Comment 3-32: Line 271. Could you please explain why cometary CAIs are younger than CM CAIs.

Reply 3-32: At least, two cometary CAIs analyzed for Al-Mg isotopes are younger than the CM-CAIs (due to remelting event(s)). But I do not think all the cometary CAIs are younger than the CM-CAIs, and we may find oldest cometary CAIs in the future.

Comment 3-33: Line 274. If CAIs were remelted during chondrule formation (rather common situation for Type C CAIs), they must be considered as chondrules formed by melting of CAI-like precursors.

Reply 3-33: The two young cometary CAIs consist of refractory minerals such as spinel, anorthite, and Al-Ti-diopside but not ferromagnesian silicate unlike chondrules (Joswiak et al., 2017 M&PS). So, they are regarded as CAIs, though they may have remelted in the chondrule forming regions.

Comment 3-34: Line 287. ... apparently ... Do you mean “appear to be”?

Reply 3-34: Changed to “appear to be” (L286).

Comment 3-35: Line 298. .. smallness ... - small sizes ...

Reply 3-35: Modified (L297).